# Modelling the role of groundwater hydro-refugia in East African hominin evolution and dispersal

M.O. Cuthbert[1,2,3,4,5], T. Gleeson[6], S.C. Reynolds[7], M.R. Bennett[7], A.C. Newton[7], C.J. McCormack[4] & G.M. Ashley[8]

Water is a fundamental resource, yet its spatiotemporal availability in East Africa is poorly understood. This is the area where most hominin first occurrences are located, and consequently the potential role of water in hominin evolution and dispersal remains unresolved. Here, we show that hundreds of springs currently distributed across East Africa could function as persistent groundwater hydro-refugia through orbital-scale climate cycles. Groundwater buffers climate variability according to spatially variable groundwater response times determined by geology and topography. Using an agent-based model, grounded on the present day landscape, we show that groundwater availability would have been critical to supporting isolated networks of hydro-refugia during dry periods when potable surface water was scarce. This may have facilitated unexpected variations in isolation and dispersal of hominin populations in the past. Our results therefore provide a new environmental framework in which to understand how patterns of taxonomic diversity in hominins may have developed.

[1] School of Earth and Ocean Sciences, Cardiff University, Main Building, Park Pl, Cardiff CF10 3AT, UK. [2] Water Research Institute, Cardiff University, The Sir Martin Evans Building, Museum Ave, Cardiff CF10 3AX, UK. [3] Department of Geography, University College London, Gower Street, London WC1E 6BT, UK. [4] School of Geography, Earth and Environmental Sciences, University of Birmingham, Birmingham B15 2TT, UK. [5] Connected Waters Initiative Research Centre, University of New South Wales, Sydney, New South Wales 2052, Australia. [6] Department of Civil Engineering and School of Earth and Ocean Sciences, University of Victoria, PO Box 1700, ECS 316, 3800 Finnery Road, Victoria, British Colombia, Canada V8W 2Y2. [7] Institute for Studies of Landscape and Human Evolution, Faculty of Science and Technology, Bournemouth University, Bournemouth BH12 5BB, UK. [8] Department of Earth & Planetary Sciences, Rutgers University, Piscataway, New Jersey 08854-8066, USA. Correspondence and requests for materials should be addressed to M.O.C. (email: cuthbertm2@cardiff.ac.uk).

Establishing the link between environmental change and resource availability in the East African Rift System (EARS) is a matter of intense debate, the resolution of which is fundamental to understanding hominin evolution as well as patterns of dispersal[1]. Most authorities currently suggest that climate change was a key factor for hominin evolution[2–4] based on the coincidence of climate shifts with major evolutionary events such as the appearance and extinctions of hominin species[5–7]. Some focus on the idea that evolutionary adaptation was caused by multiple climate cycles and enhanced pulses of climate variability[4,6]. The apparent coincidence of enhanced lake levels with evolutionary pulses is also considered significant by others[8,9]. In all cases, large-scale climatic events are held to be responsible for modification of local habitats and resource distributions causing evolutionary consequences[10], although the detailed evolutionary mechanisms remain unclear. The variability in the hydrological landscape is seen as having an important role[4,11]. However, the potential for widespread groundwater hydro-refugia, such as springs and groundwater-fed perennial streams, has long been neglected; yet it may challenge prevailing views regarding the environmental context for hominin evolution and dispersal. We use this perspective to stimulate fresh thinking around the climate-forcing hypotheses, by focusing specifically on how hydrological aspects of the landscape interact with climate change to control water availability, a key resource for survival.

In the EARS where most hominin first occurrences are located[12], potable water features in the form of surface water which are persistent on greater than seasonal timescales, are scarce. For example, lakes are often alkaline, saline and thought to have been increasingly ephemeral during the dry parts of precessionally forced climate cycles in the Plio-Pleistocene, within critical periods for hominin survival[5,13,14]. Present day conditions in much of the EARS are analogous to relatively dry periods [$\sim$70% is arid to semi-arid with groundwater recharge of $<$50 mm y$^{-1}$ (ref. 15)] and therefore provide a way of exploring the likely hydrological conditions experienced by early hominins.

Here, we show how hydrogeological modelling of the present landscape coupled with agent-based modelling of hominin movement yields new insight into potential correlates of hominin survival and dispersal. Because groundwater acts to buffer climate variability it could have provided, via springs and baseflow to perennial streams, hydro-refugia which persisted through long dry periods. In past transitions to wetter periods, trans-rift dispersal routes may have become active before those along the rift axis, and under the wettest scenarios modelled hominin dispersal (and therefore gene flow) may have been widely possible across the region. The hydro-refugia model shows that early hominins, and later *Homo*, survival and dispersal is likely to have been facilitated under drier conditions than previously thought possible.

## Results

**Present day fresh water distribution in East Africa.** Our hydrological mapping and modelling focused on groundwater manifested as springs but also assumed that major regional rivers, for which there is geological evidence of persistence through dry periods[16], were groundwater fed. Here, we therefore define both these types of hydrological features as potential groundwater hydro-refugia. We quantified the distribution and persistence of springs by first mapping existing 'permanent' springs through the eastern branch of the EARS, then modelling the temporal persistence of each spring above a flow threshold required for a spring to act as a significant water source, within an otherwise dry landscape (that is, 1,000 m$^3$ y$^{-1}$—enough running water to provide drinking requirements for hundreds of animals and to

sustain a small wetland—see Methods). In the present day, despite a likely mapping bias towards underestimation especially in the more humid and upland areas, over 450 such springs occur in the region (Fig. 1). Of these over 85% are fresh, and provide the only naturally available potable water year-round within semi-arid and arid areas. In contrast, the majority of modern lakes and streams in the eastern branch of the EARS are either alkaline/ saline or ephemeral, with the exception of catchments that drain the extensive Ethiopian Highlands, an area of above average rainfall compared to the wider region. Of the lakes that are fresh (8 out of 34), all are small, with the exception of Lake Turkana (Kenya) (Fig. 1). While brackish and relatively alkaline in the present day, this lake nevertheless supports a fresh water fish population and is therefore considered here as a 'fresh' lake, despite not being able to sustain modern humans on a sustained basis. Notably, fresh water springs often occur within the catchments of, and sometimes directly adjacent to less potable saline/alkaline lakes, such as Lake Natron (Tanzania, Fig. 1).

**Controls on the presence and persistence of springs.** Within the drier parts of the study area where spring mapping is most reliable, in comparison to what would be expected if springs were randomly distributed across the landscape, there is a slight bias for fewer springs to be located in the most arid areas (Fig. 2). However, there is also a counter-bias such that those springs modelled as being most persistent are more likely to occur in the driest areas. This runs counter to the intuition that catchments receiving more groundwater recharge will have springs which persist for longer during dry periods and yields the surprising result that modern climate is not the primary control on spring persistence in this context. This is explained by spring persistence being strongly correlated with groundwater response time, which is a function of subsurface hydraulic properties (storage and transmissivity) and geometry (length scale and topographic gradient) (see Methods, Supplementary Table 1). In general terms, the groundwater response time is a measure of how long an aquifer takes to respond to a change in boundary conditions, such as rates of groundwater recharge varying due to climate change. Therefore, given the same variations in groundwater recharge through time, spring discharge from an aquifer with a large groundwater response time will be relatively temporally stable in comparison to spring flows issuing from an aquifer with a much shorter groundwater response time. In wetter areas, while more groundwater recharge may be available to eventually discharge at springs, the resulting higher water table leads to more intersection between the water table and the topography. This reduces the distances between points of groundwater discharge, greatly reduces the value of groundwater response time, and thus increases the responsiveness of springs to variations in climate. The persistence of a spring during a dry period is thus a complex function of the timescale of climate variability, the topography which determines the catchment of the spring, and the hydraulic properties of the aquifer, all of which determine the temporal relationship between the recharge input and the spring discharge. The models we have used integrate all these factors to determine the primary controls.

Whereas previous research has focussed on climate variability being the dominant control on the availability of water, the data and models presented here show that geology and topography act to greatly buffer the impact of climate variability. For example, our results indicate that a majority of springs ($\sim$40–60%) would remain productive for periods of hundreds of years (Fig. 3) and around 30% (that is, $>$100 springs) would still remain as hydro-refugia in even drier parts of precessional cycles, assuming transitions from modern recharge conditions to conditions of just

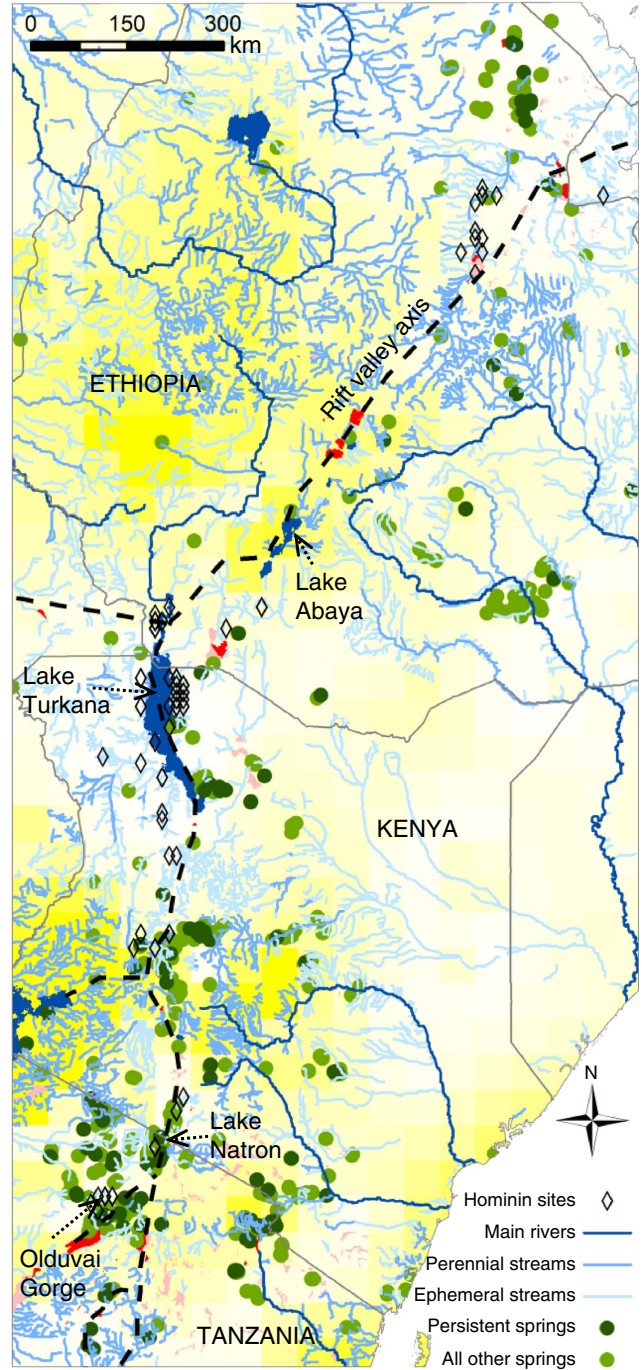

**Figure 1 | The pattern and mode of hydrologically available water in present day eastern Africa.** The distribution of water is controlled by geology, topography and climate. Hominin sites are closely associated with the rift valley axis. Fresh water lakes (dark blue), alkaline/saline lakes (red), wetlands (pink), background is groundwater recharge from Döll & Fiedler[15] coloured yellow ($250\,mm\,y^{-1}$) to white ($0\,mm\,y^{-1}$). 'Persistent Springs' are those modelled as productive ($>1,000\,m^3\,y^{-1}$) at precessional (23 ky) minima under gradual climate change. The number of such springs is considered conservative, since at least some persistent springs are likely to be present during dry periods in areas currently mapped as having perennial streams. Streams, lakes and marshes digitized from map series as described in the Methods; National borders and main rivers $>5\,km^3\,y^{-1}$ from 'Major rivers of the World, classified by mean annual discharge', GRDC http://grdc.bafg.de. Projection: WGS 1984.

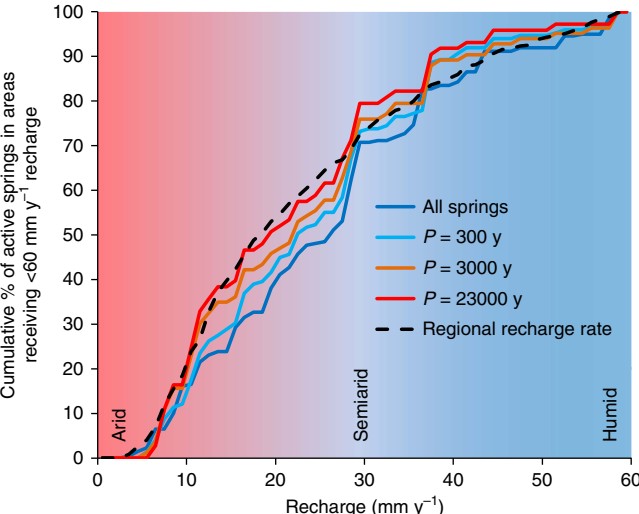

**Figure 2 | Cumulative frequency distribution of springs as a function of the average catchment recharge compared with the regional recharge distribution in areas receiving $<60\,mm\,y^{-1}$ recharge.** Springs were considered as 'active' if they maintain flow $>1,000\,m^3\,y^{-1}$ for gradual recharge variations sinusoidally fluctuating from zero to the average catchment recharge with period (P). The 'regional recharge rate' curve is indicative of the cumulative frequency distribution that would be expected if springs were randomly distributed across the landscape. The deviations between the actual spring distributions indicate that spring persistence (as opposed to spring presence) is not controlled primarily by modern climate, consistent with statistical results which show that spring persistence is not significantly correlated to any individual catchment characteristic including groundwater recharge (Supplementary Table 1).

$1\,mm\,y^{-1}$ recharge (that is, arid-hyperarid). For a sudden hydrological transition to arid conditions, we see an initially quicker decline in the number of springs still remaining productive, as would be expected (Fig. 3 and Supplementary Fig. 1). Sensitivity analysis and Monte Carlo experiments indicate that while the hydraulic parameter uncertainty for our estimates of spring persistence for an average spring is ±37%, the maximum error in the combined modelled percentage of springs persistent on any timescale is only ±5% (Fig. 4, Supplementary Figs 2 and 3).

**The abundance of persistent groundwater hydro-refugia.** There are only a small number of hydro-refugia which are known from the geological record to have survived during the driest periods of precessional climate cycles and are also associated with hominin fossils and stone tools. Lake Turkana *c.* 2–1.85 My[16] (Kenya, Fig. 1) is an important example, supported by the paleo-River Omo, which has a catchment in the Ethiopian Highlands. Although there is an increasing recognition of paleo-springs in the geological record, they are still relatively rare. This is likely due to their low preservation potential; groundwater discharge only leaves a direct geological record under specific geochemical conditions that lead to mineral precipitation (such as tufa[17]). However, a spring located at paleo- Lake Olduvai *c.* 1.84 My[17] (Tanzania, Fig. 1) is thought to have continued to flow during a prolonged period of aridity[18] at the precessional minimum. Our results suggest that there would have been orders of magnitude more groundwater hydro-refugia during arid phases than the geological record currently indicates. Furthermore, groundwater hydro-refugia would have been much more abundant than fresh water potable lakes during periods of prolonged aridity. Since some recharge still occurs even in arid areas in the present

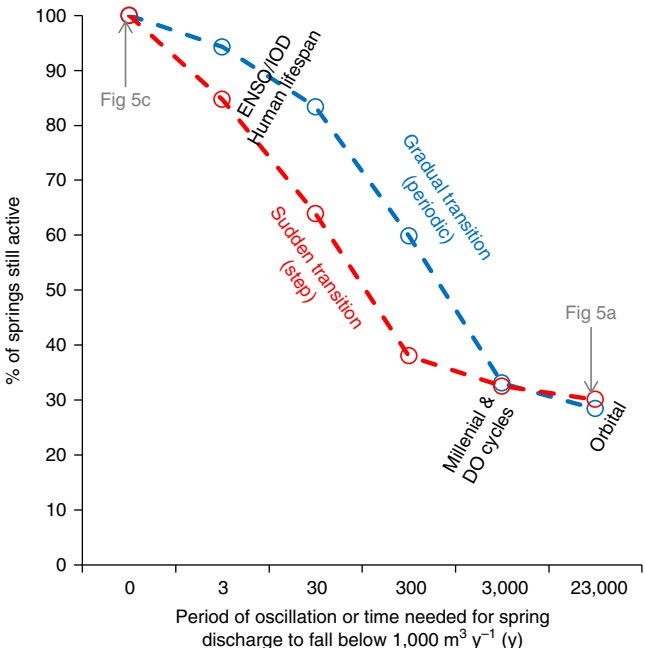

**Figure 3 | Many groundwater hydro-refugia persist through even precessional orbital climate cycles.** Modelled frequency of springs that are still active (that is, flow rate still above 1,000 m³ y⁻¹) at the driest part of climate changes over a range of timescales under gradual or sudden boundary condition changes (ENSO—El Nino—Southern Oscillation; IOD—Indian Ocean Dipole; DO—Dansgaard–Oeschger). Recharge is assumed to vary between the modern day value to a minimum value of 1 mm y⁻¹. Sensitivity of persistence to the recharge occurring at the driest point in a climate cycle is explored in Supplementary Fig. 1.

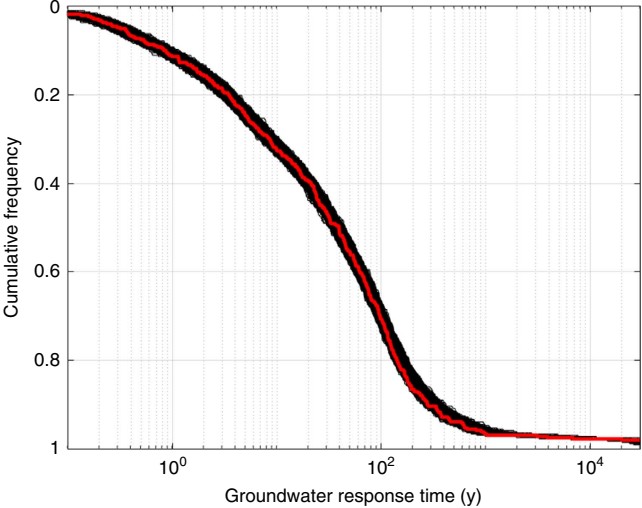

**Figure 4 | Cumulative frequency distribution of groundwater response times for 10,000 Monte Carlo realizations indicates that the combined parameter uncertainty in the CDF of modelled spring longevity across the EARS is ± 5%.** The result using an average set of parameters (plotted in red) is superimposed on all other realizations (plotted in black). The maximum spread occurs at the steepest part of the curve for groundwater response times of 100–200 years. Since groundwater response times vary by five orders of magnitude, changes in groundwater response times can have more impact on spring persistence than changes in recharge. However, only a small amount of recharge is needed to maintain flow to a spring; for example, a recharge of just 1 mm y⁻¹ over an area of 1 km² could maintain flows of 1,000 m³ y⁻¹. Therefore, while a small amount of recharge is a necessary, but not always sufficient, condition for a persistent spring, the geological and topographical characteristics of the EARS are also fundamental controls.

day[15,19–21], the 1 mm y⁻¹ recharge applied during the driest periods used in our models leads to a conservatively low number of predicted active springs during these periods. The fact that at least some persistent springs are likely to have been present during past dry periods in areas mapped as having perennial streams in the present day also suggests our predictions are conservative (Supplementary Fig. 4). Nevertheless, while the presence of spring-based refugia has been demonstrated for an isolated site[22,23] our results suggest their presence and importance over a much wider geographical area.

**Modelling the dispersal of hominins.** We examine the evolutionary implications of this improved hydrological understanding using an agent-based model (ABM). The specific advantages of using an ABM in this type of context are summarized by Bonabeau[24] who identifies three particular benefits compared to other modelling approaches, namely their ability to: (i) capture emergent phenomena; (ii) provide a natural description of a system; and (iii) be relatively flexible. We apply the principal of uniformitarianism and consequently use the contemporary landscape as a method of exploring potential constraints on hominin movement imposed by the availability of water (that is, springs, streams, lakes or wetlands). Our aim is to understand the role that changes in the generic hydrological resource network, through a simulated climate cycle, may have played as one control on hominin movement. Here, we do not use the ABM to examine explicit hypothesizes related to past events or scenarios, rather we examine generic principles that might apply in this and past landscapes. The connectivity between two water sources in our study does not depend on fixed properties based on their location, but arises as an emergent property of the simulated

system's constituent units (the agents) and their interactions with the terrain. The landscape is heterogeneous in terms of land cover, slope and roughness, and this affects the ability of agents to traverse an area. This is explicitly incorporated by including variation in transit time in response to terrain properties within the model. There are many potential routes between two points in a landscape, and our model allows agents to explore a wide range of different routes, according to the decisions that would be made by an individual walking across a landscape. Only those successful transits between water bodies are considered in our analysis as evidence of linkage between 'nodes' (or water bodies).

We have modelled a range of climate scenarios along the continuum of a precessional orbital climate cycle. In the absence of specific data on relative spatial changes in effective precipitation through a precessional cycle (for example, regionally, or say between highlands and lowlands), it is assumed that as the climate becomes more arid, decreases in effective rainfall and groundwater recharge occur proportionally everywhere leading to progressive shrinking and fragmentation of the hydrological network. Conversely as rainfall/recharge increases, we assume that water tables increasingly intersect stream and lake beds and there is enhanced potential for springs to occur at higher elevations, all of which leads to an expanded hydrological network. We recognize four hydrological components in our hominin mobility model: (1) mapped springs (seasonal, perennial and geothermal); (2) mapped streams (seasonal or perennial), major rivers using flow thresholds set at >0 or >5 km³ y⁻¹ (GRDC data, see Methods); (3) mapped wetlands (seasonal, perennial); and (4) mapped lakes (fresh, saline and seasonal). Four hydrological scenarios were run representing a dry to wet

| Run Id | Climate state | Hydrological components |
|---|---|---|
| Run-1 | Future wet | Modern perennial/seasonal springs, geothermal springs, fresh water and saline lakes, perennial/seasonal rivers, major rivers (flow $>0\,km^3\,y^{-1}$) and perennial/seasonal wetland/marsh. |
| Run-2 | Present wet | Modern perennial springs, geothermal springs, fresh water and saline lakes, perennial rivers, major rivers (flow $>0\,km^3\,y^{-1}$) and perennial wetland/marsh. |
| Run-3 | Present dry | Modern perennial springs, geothermal springs, fresh water and saline lakes, major rivers (flow $>0\,km^3\,y^{-1}$) and perennial wetland/marsh. |
| Run-4 | Future dry | Persistent springs (that is, those modelled as being persistent through precessional cycles), geothermal springs, fresh water lakes and major rivers (flow $>5\,km^3\,y^{-1}$) with saline lakes excluded as either desiccated or too hyper-saline to be potable. |

**Table 1 | Climatic scenarios modelled using the agent-based model.**

continuum (Table 1). Run-4 corresponds to the driest phases of a precessional climate cycle (for example, 23 ky), where predominantly arid conditions existed across eastern Africa, with only the most persistent springs, deepest lakes and highest order and/or groundwater-fed sections of the stream network present.

**Controls on hominin mobility and gene flow.** The modelling results demonstrate: (1) That some hominin movement may have been possible between spring networks and along major rivers (groundwater hydro-refugia) that would have allowed hominin populations to survive in specific regions, even during the most extreme arid climate phases (Figs 5a and 6a). The presence of hydro-refugia during the driest of scenarios is robust taking into account uncertainty in the model input parameters (Supplementary Fig. 6). (2) That under the 'present dry' conditions modelled hominin mobility occurs transverse to the rift axis rather than along it (Figs 5b and 6b). In specific cases modelled, modern springs high on the rift margins play a role in connecting the rift floor hydrological system with those of the rift flanks. Note the presence of cross-rift connections in the Ethiopian Highlands is sensitive to the input parameters used in the model (Supplementary Fig. 7) but still holds for other areas of the rift (Fig. 5b and Supplementary Fig. 8). The importance of dispersal routes transverse to the rift axis is at odds with the common assumption of along-rift dispersal[23,25] but shows agreement with westward dispersals observed in some hominins[26] and genetic studies of several other species[27,28]. (3) Under the wetter scenarios modelled (Figs 5c and 6c and Supplementary Fig. 8), in which the fluvial network becomes dominant, the potential for widespread hominin dispersal as well as associated gene flow is evident. These conclusions are summarized in Fig. 7 and have been found to be statistically robust through model repetition (see Methods; Supplementary Table 2 and Supplementary Fig. 15).

In addition to the long timescale changes in climate expected through a precessional cycle, shorter term variations (for example, seasonal dry periods or multi-year droughts) would have altered the availability of fresh water. For the 'present' scenarios, the effect of seasonality on the potential connectivity of hydro-refugia is incorporated in the analysis by comparing the 'present wet' and 'present dry' scenarios (Fig. 5b and Supplementary Fig. 10). Such patterns of variation in the location of available spring water are in accordance with the modern experience of East African communities[29,30]. For the 'future wet' scenario, mobility is already so easy that seasonality would make little difference (Fig. 5c). For the 'future dry' scenario, the impact of seasonality is harder to constrain but in the driest parts of the precessional cycle envisaged, seasonal expansion of the drainage network is likely to have been much less than that during the present day. This is because both runoff and recharge are strongly controlled by antecedent moisture and water table conditions[21,31]. Hence while seasonal mobility may be enhanced to some extent even during short wet periods, a sustained drier prevailing climate would result in decreased streamflow and a less expansive stream network than observed in the present day even during periods of relative (for example, seasonal) wetness.

## Discussion
One of the fundamental problems in explaining human evolution by invoking climate in East Africa is that precessional climate forcing occurs on a timescale that appears too short for allopatric speciation to occur. To counter this, one has to invoke the variability itself as the key factor[4] or look to longer eccentricity cycles and their modulation of the precessional cycle amplitude[5]. Our work suggests three alternative possibilities, however: (1) a different geographical distribution of landscape elements at various times in the past may have favoured longer term periods of isolation; (2) that the population density was such that while connectivity was possible it was not exploited; or, (3) more likely in our view, that climate may not play such a primary role in human evolution, as is commonly asserted. In fact, the potential for frequent, widespread dispersal as illustrated here, and the resulting potential episodes of genetic admixtures, might explain the lack of phylogenetic diversity in the hominin lineage noted by some[32].

In conclusion, the hydro-refugia model therefore points to the need to evaluate a range of parameters and variables beyond the current paradigm of climate-driven environmental change to explain hominin evolution. It provides a basis for palaeoanthropology to explore the possible mechanisms by which taxonomic diversity in hominins arose[32], and may help to explain levels of genetic exchange identified in ancient African populations[33]. Our hydrological results have wide global applicability (drylands cover around 45% of the Earth's landmass[34]) and the importance of groundwater for the survival of our hominin ancestors when faced with dramatic climate changes in the past could also inspire and inform strategies for human resilience to future climate change[35,36].

## Methods
**Data compilation and hydrological mapping.** The study covers an area of 2,093,280 km² stretching from northern Tanzania ($-5.390°$S, $33.996°$E) to Ethiopia ($13.797°$N, $42.799°$E) and focused along the eastern African Rift.

Taxonomic occurrences of early (pre *Homo sapiens*) Hominidae in Ethiopia, Kenya and Tanzania, representing 178 sites in total, were collated from all available records in the Paleobiology Database (Fossilworks: http://fossilworks.org), and their co-ordinates digitized.

Stream locations were digitized from 1:500,000 British Army maps dating from the 1940s with gaps in-filled from: (1) modern 1:250,000 topographic maps, Kenyan Government, 1981; (2) East Africa, Series 1501, Joint Operations Graphic (Air) 1:250,000 US Air Force; or (3) East Africa 1:250,000, Series Y503, British Overseas Mapping circa 1963 (http://www.lib.utexas.edu/maps/ams/east_africa). Second-order rivers were digitized for both seasonal and non-seasonal cases. Modelled flow data was used to define major perennial rivers as published by GRDC[37], based on WaterGap 2.1 global hydrological model output. Lake outlines were obtained from Digital Chart of the World (http://www.diva-gis.org/gdata) with salinity for named lakes determined via literature. If salinity information was not available then the lake was classed as 'fresh' by default.

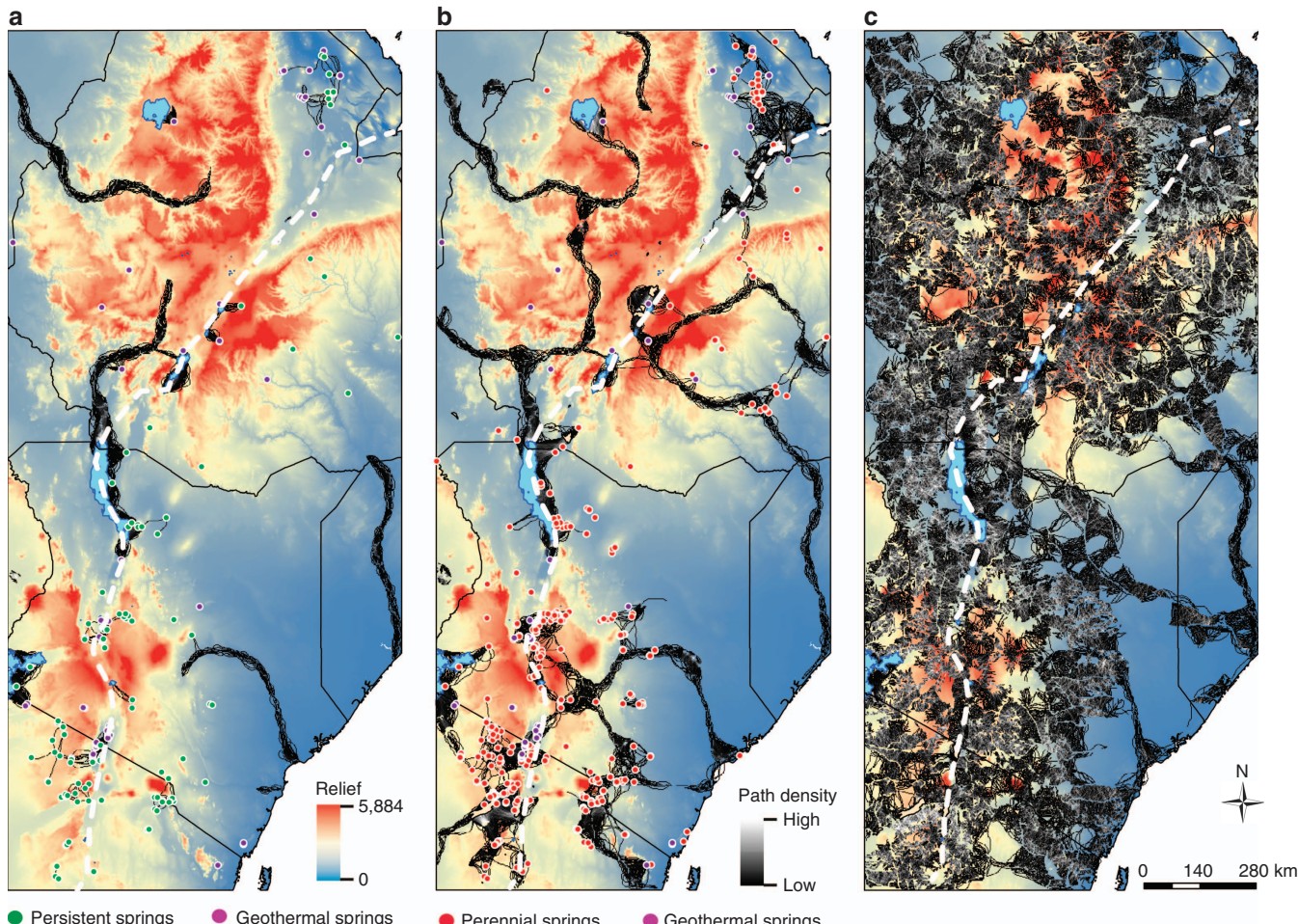

**Figure 5 | Modelled dispersal of hominins between hydro-refugia through a wet–dry climate cycle.** As the climate changes from the driest (**a**) to wettest (**c**) state envisaged during a precessional cycle, the modelled dispersal of hominins between hydro-refugia increases until dispersal is almost ubiquitous. Agent-based modelling results based on three hydrological scenarios, using a maximum three-day travel distance of 150 km and surface roughness as the cost layer scaled according to Supplementary Fig. 11. The black lines shown represent the tracks of agents in the model. It is important to note that just because the model shows a potential pathway between two water sources it does not necessarily mean that it was actually followed. (**a**) Driest scenario using persistent springs (Fig. 1), geothermal springs, fresh water lakes, major rivers with a flow $> 5 \, km^3 \, y^{-1}$ (Run-4). Note the networks of springs acting as hydro-refugia, which persist irrespective of the topographic cost layer and scaling used in the model (Supplementary Figs 9–12). (**b**) Dry scenario using modern springs (perennial + geothermal), lakes (fresh + saline), perennial wetlands and major rivers with a flow $> 0 \, km^3 \, y^{-1}$ (Run-3). Note the potential for dispersal transverse to the rift axis (white dashed line) and the absence of along-axis dispersal routes. Springs act to connect rivers on the rift flank with those of the rift floor. The cross-rift movement in the vicinity of Lake Abaya (Ethiopia), where the rift cuts the Ethiopian Highlands, is sensitive to the topographic cost layer and scaling used in the model (Supplementary Fig. 9); however, the principle holds in southern Kenya. The next stage in the increased water availability continuum is shown in Supplementary Fig. 10, which includes perennial streams and reveals a progressive increase in cross-rift movement and the start of dispersal along the rift (Run-2). (**c**) Wettest scenario uses modern springs (seasonal, perennial + geothermal), wetlands (perennial + seasonal), lakes (fresh + saline), major rivers with a flow $> 0 \, km^3 \, y^{-1}$ (Run-1). Note the potential for widespread dispersal of hominins and genes. Background relief map is based on a $30 \times 30$ m SRTM model: http://earthexplorer.usgs.gov/. Projection: WGS 1984.

The location of the main East African Rift Valley axes were digitized from Hayes *et al.*[38] Locations of springs were digitized from a variety of available map sources to enable 100% coverage for the study area on as consistent a basis as possible as follows: Ethiopia (East Africa Y401—GSGS 4355, 1:500,000 scale), Kenya (East Africa JOG 1501 AIR, Y503, 1:250,000 scale; Tanzania (East Africa Y401—GSGS 4355, 1:500,000 scale and Geological Map 1st Edition Quarter Degree Sheets, 1:125,000 scale). Mapping at smaller scales inevitably leads to a larger number of mapped springs. A comparison of map sheets from Tanzania where we have two scales of maps to compare indicates that the 1:500,000 maps record, on average, 40% of the number of springs which are present on the 1:125,000 maps. The 1:125,000 maps have been 'ground-truthed' by the authors (unpublished) across several map sheets in northern Tanzania with the input of local Masai guides, suggesting they are remarkably accurate in representing the main sources of water used by local people. However, the scale of mapping across the modelled area generally decreased in resolution from 1:125,000 in Tanzania, to 1:250,000 in Kenya, and 1:500,000 in Ethiopia. Thus, we expect that the undersampling due to the changing map scales leads to a loss of accuracy in the absolute spring count from close to 100% accuracy in the south of the area to around 40% in the north.

We are also aware of possible bias of maps underestimating the number of springs in the wettest areas of the study area, where perennial streams are frequent and springs may thus be overlooked. This is consistent with known springs used for water supply in the Ethiopian Highlands (for example, Calow *et al.*[29]) which were not represented on the large-scale maps. Given the likely underestimation of the total number of springs, to ensure the sample of springs we have mapped is nevertheless representative for making inferences regarding spring persistence, we have randomly resampled the total set of springs for increasing subsample sizes as a proportion of the total spring set. For each subsample, we calculated the RMSE for the cumulative frequency distribution of groundwater response times of the subsample against the full spring set. The plot of RMSE versus proportion of the total sample size (Supplementary Fig. 4) demonstrates that the set of springs we have mapped is representative as the errors reduce to zero for a subsample size of around 75% of the full set.

ArcGIS was used to delineate for each spring: the distance to the nearest watershed boundary (*B*) average slope (α) using the Hydrosheds database[39] and a 90 m spatial resolution Digital Elevation Model[40]. The Hydrosheds database was also used to derive upstream contributing catchment areas (*A*). Hydraulic

conductivity ($k_0$) and porosity ($n_e$) values for each spring location were sourced from Gleeson et al.[41], which is based on the highest resolution mapping digitally available[42]. While, we recognize that there may be local variations in groundwater recharge[43–45], we used the distribution of potential groundwater recharge from Döll and Fiedler[15] consistent with other regional African groundwater studies[42]. The range of input parameter values are plotted in Supplementary Fig. 2.

ArcGIS was used to analyse relationships between spring persistence, climate conditions and catchment properties.

**Development of models of spring persistence.** East Africa is a very diverse but under-researched hydrogeological environment. There are also inherent

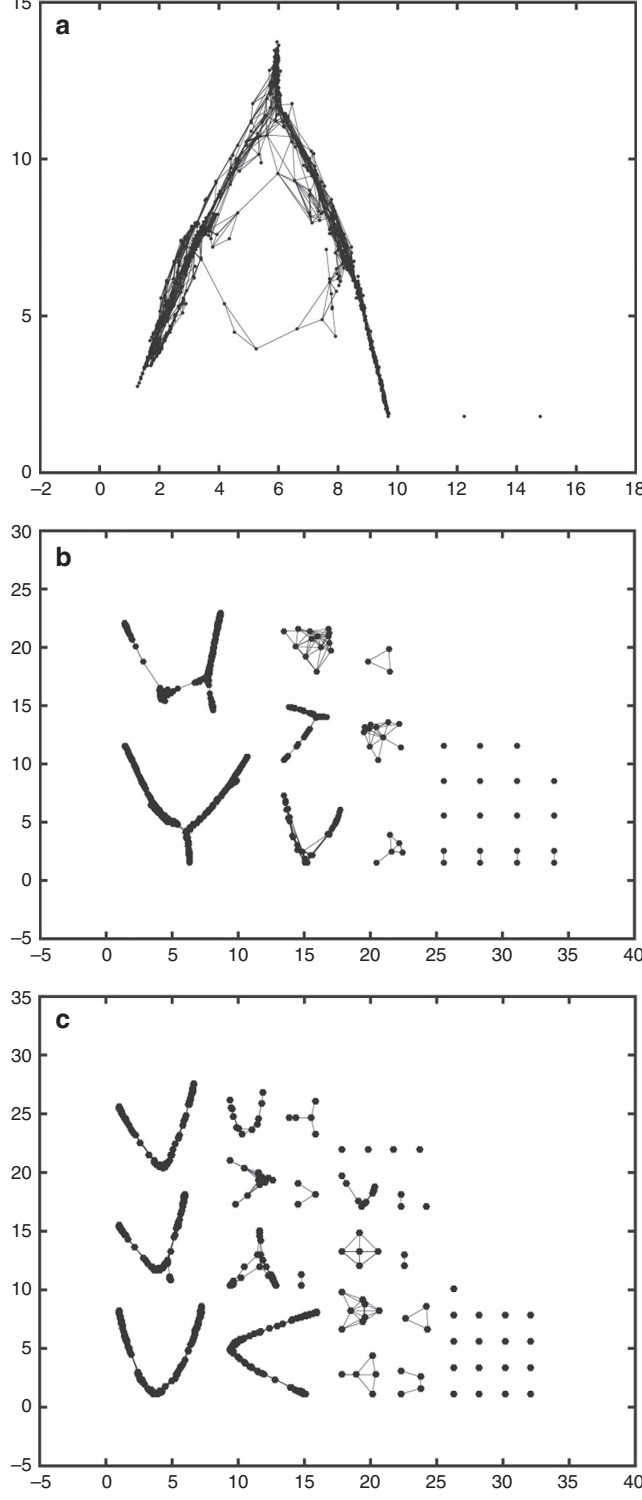

uncertainties involved in modelling long timescale climatic changes in this context. Thus our choice of modelling approach needed to be generic enough to cover the most important features of groundwater flow systems across a range of contexts in which EARS springs are found, while being mathematically simple enough to apply existing analytical solutions to the governing flow equations to enable the uncertainties to be easily explored. Although there is debate about the distribution and precise role of transverse faulting in controlling groundwater flow in the EARS[46], an emerging pattern is one of nested flow patterns whereby groundwater age increases from rift flank to graben[46–48]. Due to the heterogeneity and anisotropy of many EARS lithologies which in some locations determines spring locations, it would be virtually impossible to predict the locations of all springs on the basis of regional scale geological mapping and analytical flow models alone. However, by mapping the locations of the present day springs directly, it is then reasonable to apply a simple flow model and bulk hydraulic parameters to determine the likely variability of the springs to climate variations.

Each spring was therefore modelled using a one-dimensional (1D) linearized Boussinnesq equation as follows:

$$\frac{\partial \eta}{\partial t} = \frac{k_0 \eta_0 \cos \alpha}{n_e} \frac{\partial^2 \eta}{\partial x^2} + \frac{k_0 \sin \alpha}{n_e} \frac{\partial \eta}{\partial x}, \tag{1}$$

where $t$ is time [T], $x$ is distance along the aquifer base [L], $\eta$ is hydraulic head [L], $k_0$ is aquifer hydraulic conductivity, $n_e$ is aquifer specific yield [ − ], $\alpha$ is the slope of the aquifer [ − ], $\eta_0$ is the average water table height equal to $pD$, where $D$ is the maximum saturated thickness of the aquifer [L] and $p$ is a linearization constant normally assumed to equal 0.3.

This approach assumes that a homogeneous, isotropic, sloping aquifer extending to a watershed boundary at one end (no flow boundary condition at $x = B_x = B/\cos(\alpha)$) receives groundwater recharge uniformly over its surface and transmits groundwater discharge to a spring at its lower end (Dirichlet boundary condition at $x = 0$). In reality, preferential flow through high-permeability fracture features may dominate hydraulic head distributions at a local scale in some spring systems. However, it has been shown that even in such cases, similarly idealized analytical spring flow models to those used here can correctly simulate the observed spring discharge dynamics of the bulk groundwater flow system[49]. Our models neglect heterogeneity and anisotropy and assume isotropic hydraulic parameters—hence, they could be improved in the future if higher resolution maps of the required hydrogeological parameters become available for the region. Two scenarios were modelled for 'sudden' and 'gradual' climate change end members as follows.

First, models for sudden climate change were implemented assuming recharge ($R$) ceases entirely after a period of climatic steady state. Following Brutsaert[50], spring discharge ($q$) is given by:

$$q(t) = -2B_x R \cos \alpha \sum_{n=1,2,3...}^{\infty} \frac{z_n^2 \left[1 - 2\cos(z_n)\exp\left(\frac{Hi}{2}\right)\right] \exp\left[-\left(z_n^2 + \frac{Hi^2}{4}\right)t_+\right]}{\left(z_n^2 + \frac{Hi^2}{4} + \frac{Hi}{2}\right)\left(z_n^2 + \frac{Hi^2}{4}\right)}, \tag{2}$$

with $z_n$ being the $n$th root of $\tan(z) = -2z/Hi$ and:

$$Hi = \frac{B_x \tan \alpha}{\eta_0}, \tag{3}$$

$$t_+ = \left[\frac{k_0 \eta_0 \cos \alpha}{n_e B_x^2}\right] t. \tag{4}$$

Second, models for gradual climate change were implemented using sinusoidally varying recharge as a top boundary condition as follows:

$$R = R_{av}(1 - \cos(\omega t)), \tag{5}$$

with recharge thus varying between zero (that is, hyperarid conditions) and a maximum value (that is, $2R_{av}$) for periods ($P = 2\pi/\omega$) ranging from 1 y to 23 ky.

**Figure 6 | Network shapes derived from successful journey matrices output from the PATH model across the modelled climatic cycle based on ten repeated runs.** (**a**) Future wet scenario (Run-1). (**b**) Present dry scenario (Run-3). (**c**) Dry (23 ka) scenario (Run-4). Note the progressive increase in the number of sub-networks. The axes are dimensionless and are expressed in non-geographic units. The networks simply portray the relationship between nodes linked by common edges (that is, one or more successful journey). Each sub-graph represents a distinct network that is unlinked to any other by an edge; the more the sub-graphs, the poorer the connectivity between the sum of the nodes present. The networks are derived from matrix of successful/unsuccessful journeys and were created by exporting the successful journey matrix from the PATH model and plotting nodes and edges within Matlab.

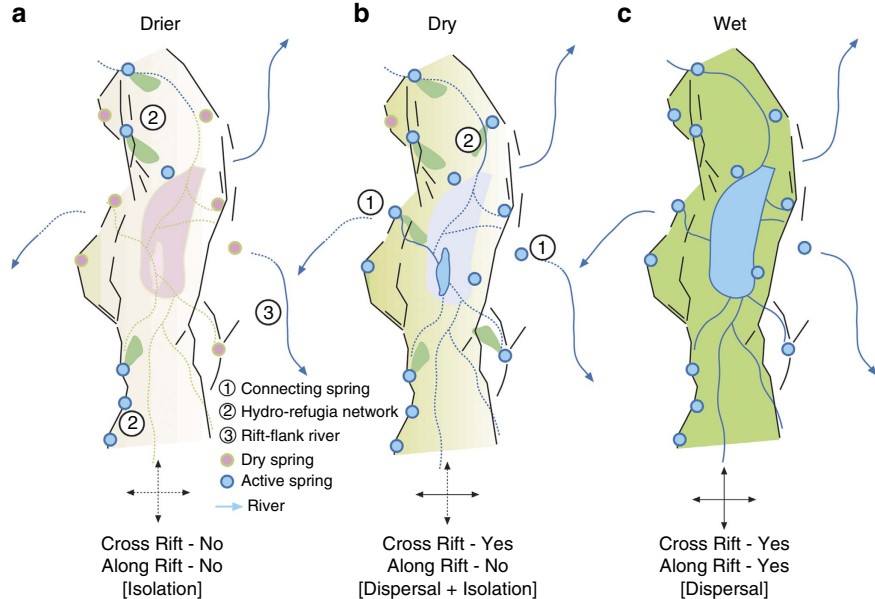

**Figure 7 | Conceptual model showing the role of springs across various climate scenarios.** Under the drier scenario (**a**) hominin survival is focused on single springs (or spring clusters) and movement between springs (or spring clusters) is limited. As climate improves (**b**) the availability of water sources increases particularly as the water table intersects rift flank rivers. Springs high on the rift sides may act to link rift flank rivers with water sources in the rift facilitating transverse rift movement. As water becomes widely available (**c**) hominin movement occurs in all directions including along the rift axis.

The solution for this case was derived by Cuthbert & Ashley[22] as:

$$q(t) = R_{av} \sum_{n=1,2,3...}^{\infty} \left[ \frac{A}{C^2 + \omega^2} \left( -\omega \sin(\omega t) + C \cos(\omega t) \right) - \frac{A}{C} \right], \quad (6)$$

$$A = -\frac{2k_0 \eta_0 \cos \alpha}{n_e B_x} \left[ \frac{z_n^2 \left[ 1 - 2 \cos(z_n) \exp\left(\frac{Hi}{2}\right) \right]}{\left( z_n^2 + \frac{Hi^2}{4} + \frac{Hi}{2} \right)} \right], \quad (7)$$

$$C = -\frac{k_0 \eta_0 \cos \alpha}{n_e B_x^2} \left( z_n^2 + \frac{Hi^2}{4} \right). \quad (8)$$

We implemented these equations in MATLAB setting the necessary numerical tolerances to yield an error of $<1\%$ in the resulting spring persistence metrics we output. The models assume a 1D geometry with the output, $q$, being a discharge per unit width of aquifer $[L^2 T^{-1}]$. We have therefore rescaled the results to approximate the actual spring flow ($Q$, $[L^3 T^{-1}]$) for each entire spring watershed whereby $Q = A^*q/B$. In reality, the catchment geometry will often include non-uniform (predominantly convergent) flow fields, which will affect the groundwater response times as outlined below.

Available recharge values from Döll and Fiedler[15] were derived from a global hydrological model which takes no account of the underlying geology. It was therefore important to ascertain whether this recharge could be accommodated by the groundwater flow systems feeding each spring by ensuring that the modelled water table stays below the ground surface. To do this a steady-state solution to equation 1 was used for the previously stated boundary conditions as follows[51]:

$$\eta = RpD \cos(\alpha) \left\{ 1 - e^{\frac{-\tan(\alpha)x}{pD}} - \frac{\tan(i)}{pD} \left[ \left( 1 - e^{\frac{-\tan(\alpha)x}{pD}} \right) B_x + x \right] \right\} \left[ k \sin^2(\alpha) \right]^{-1}. \quad (9)$$

Further, the position of the maximum water table height is given as follows:

$$x_{max} = \frac{pD \ln\left( 1 + \frac{\tan(\alpha)L}{pD} \right)}{\tan(\alpha)} \quad (10)$$

These equations were used to calculate a refined estimate of the recharge values needed to ensure physically realistic water table conditions. These refined values were less than or equal to those derived by the global hydrological model.

The gradual and sudden end member climate scenarios were both modelled using a range of maximum recharge values including the modern day distribution of calculated actual recharge, as well as with spatially constant values from the current regional potential recharge average ($49 \, mm \, y^{-1}$). Recharge minima were defined as a $1 \, mm \, y^{-1}$ rate considered to be a conservative value for arid conditions [15,19–21], and scenarios with minima of $0 \, mm \, y^{-1}$ recharge were also run as the most extreme possible end member. We assigned hydraulic properties to the model based on the results from mapping as described above, assuming $D = 100 \, m$ and that specific yield was equal to drainable porosity, and tested the parameter uncertainty as described below.

Spring persistence was defined for the gradual end member as the maximum period of variation for which spring discharge did not fall below a threshold value,

and for the sudden end member as the time taken for spring flow to fall below this threshold following the step reduction of recharge from steady-state conditions. The spring discharge threshold was defined, conservatively, as $1,000 \, m^3 \, y^{-1}$. For example, this could be envisaged as enough running water to provide drinking requirements for 100s of animals of $\sim 2,000 \, l \, d^{-1}$ ($\sim 750 \, m^3 \, y^{-1}$) with the rest of the discharge sustaining a small wetland of $\sim 100 \, m^2$ transpiring at an average annual rate of $\sim 2,500 \, mm \, y^{-1}$ typical of (semi)arid environments ($= 250 \, m^3 \, y^{-1}$).

Springs likely appeared or disappeared due to local geological or topographic changes, such as faulting and volcanism, or blockage of groundwater discharge by sedimentation or accumulation of spring precipitates. However, we are examining the combined regional distribution of springs through time and space in the EARS, a region which is still undergoing extensional tectonics that began $\sim 30 \, Ma$ ago[52]. It has therefore, throughout the time period in which hominins evolved, provided a suitable hydrological setting by creating the necessary relief and landscapes for developing active groundwater flow systems; topographic highs that trap moisture to provide groundwater recharge and drive groundwater flow by gravity to discharge in topographic lows[53].

**Testing models of spring persistence.** It was not possible to calibrate the models directly across the region due to the sparsity of available spring flow time-series data in East Africa and the long timescales considered in this paper. However, using what data were available, a range of model tests were carried out over timescales from years to millennia as follows. One published multi-decadal spring flow time series was found within the study area (Mzima Springs, Kenya[54]) and digitized along with the next nearest available long record from South Africa (Uitenhage Spring[55]) with a comparable fractured-rock hydrogeology. For these sites, the rainfall time series (published with the spring flow records) were characterized using a superposition of two sinusoids whose relative amplitudes were used as input data to the periodic model equations (6–8) to generate spring flow output. For Mzima, hydraulic parameters were used directly from the GIS mapping described above, and for Uitenhage, the parameters were set using the published range of literature values for that site. The results for both springs show that the model performs well with respect to simulating the observed degree of lag and attenuation between the rainfall and spring flow (Supplementary Fig. 5) with a slight tendency for over-responsiveness in the model. The total spring flow for Mzima is underestimated by nearly 30%, but since it is based on uncalibrated parameters defined on the basis of regional GIS mapping, this is a very reasonable result.

The only way of directly testing the model validity over much longer timescales is through geological evidence, and output from an identical analytical model has previously been shown to be consistent with the geological record at Olduvai Gorge, Tanzania within the study area[22]. However, another approach is to compare the analytical model used here with a more complex numerical model simulation operating over longer timescales, as a proxy for actual data. Such a model developed for North Africa[56] was successfully calibrated to present day data and then hindcasted to simulate a 20 ka period of groundwater discharge recession with

no groundwater recharge since the last 'pluvial' period. Our analytical cessation equations (2–4) have therefore been applied using the range of parameters from the published 'base case' model and the outputs compared (Supplementary Fig. 5). The recession from the published study has an initially steeper recession than the analytical model, but yields a similar overall recession on the 20 ka timescale modelled.

Despite the lack of data for calibration, these tests thus demonstrate the reasonableness of modelling approach applied for the aims of quantifying, to within an order of magnitude accuracy, the response times of springs in East Africa over a range of timescales from years to millennia (Supplementary Fig. 5).

**Groundwater response times and uniform flow field assumption.** In general terms, the groundwater response time (GRT) is a measure of how long an aquifer takes to respond to a change in boundary conditions, such as rates of groundwater recharge varying due to climate change. It is given by the equation $GRT = B^2/D$ with hydraulic diffusivity for a sloping aquifer, $D = T/(n_e \cos(\alpha))$ (ref. 50), and where $T$ is the aquifer transmissivity. Our models indicate a good linear relationship ($R^2 = 0.64$, Supplementary Table 1) between GRT and spring flow recession timescales as expected. However, we note that GRTs for non-uniform flow fields increase for convergent and decrease for divergent flow geometries[57]. Thus our results, which assume uniform flow, are conservative with respect to the modelled timescales of spring persistence.

**Monte Carlo experiments and groundwater model sensitivity.** Sensitivity of modelled spring persistence to parameter uncertainty was tested by varying each parameter by $\pm 25\%$ in turn and interrogating the model output (Fig. 4). Monte Carlo experiments (MCEs) were prohibitively computationally expensive to run on the full models. However, we felt it important to explore the full range of parameter uncertainty. Thus, we used the result that spring longevity was well correlated with GRT, and ran the MCE on GRT for each spring for 10,000 combinations of parameters sampled randomly from the range defined as follows. The range of parameter uncertainty was assumed to be well constrained for mapped parameters (catchment length, area and slope) and allowed to vary with a standard deviation equal to 10% of the mean. Hydraulic properties $k_0$ and $n_e$ were free to vary through one standard deviation as defined by Gleeson et al.[41] The saturated flow thickness was deemed to be more uncertain and free to vary through a normal distribution with a standard deviation of 25% of the mean (Supplementary Fig. 3).

**Agent-based modelling.** ABM was implemented using the Pathway Analysis Through Habitat (PATH) algorithm run in Netlogo[58,59]. A copy of the original model code is available from: http://extras.springer.com/2012/978-1-4614-1256-4 and a copy of the modified code is included in Supplementary Information. The algorithm involves launching agents from defined habitat patches, in this case water, to simulate the journey made by individuals through a landscape until they either arrive at another suitable habitat patch or die. There are four model requirements: (1) a value for total travel distance in a unit of time; (2) a map of water-patches; (3) a map showing the energetic cost associated with the terrain travelled; and (4) a map showing the potential for death (lethality) associated with the terrain travelled. While the original PATH model has a capacity to include a spatially variable measure of lethality, in the interests of parsimony, we have chosen to set this to zero. This gives the agents the maximum chance of reaching their destination and achieving dispersal and thereby tests rigorously the potential for population isolation. Furthermore, sensitivity tests using a range of possible lethality do not materially alter the potential for dispersal reported here. Modelling was to a resolution of 1 km, with relief roughness[60] used as the cost layer, scaled to reflect the variation in walking speed with slope/roughness as defined by Naismith's Rule[61] (Supplementary Fig. 9a). A maximum travel time of 3 days without water was used[62]. Daily walking distances were based on modern human walking speeds. Derived from on a large sample ($N = 3,500$) Tarawneh[63] quotes pedestrian walking speeds for different age categories of the order of $< 20$ y, $1.29\,m\,s^{-1}$; $21$–$30$ y, $1.49\,m\,s^{-1}$; $31$–$41$ y, $1.47\,m\,s^{-1}$; and $> 65$ y, $1.1\,m\,s^{-1}$. While modern humans are capable of walking at speeds upwards of $2.5\,m\,s^{-1}$, especially for short distances, they typically choose to walk at their 'preferred walking speed' which is variously defined as $1.3$ and $1.4\,m\,s^{-1}$ (refs 64–67). We acknowledge that there may be other aspects of terrain, which might impede movement such as vegetation type and surface albedo or 'going characteristics', but on the basis of the range of data, one can assume that daily walking distance for a modern human is likely to vary between 40 and 50 km over a 10 h period. Taking the least conservative value, the model scenarios were run with a 3 days distance of 150 km representing a maximum possible travel distance. Values are likely to have been less for pre-*Homo* agents and a hominin troop may have moved at the pace of the slowest member. All conclusions were tested specifically against travel distances of 120, 150 and 180 km (Supplementary Table 5, Supplementary Fig. 6). While lower travel distances of say 80 km or less in 3 days are conceivable, they simply reduce the connectivity between points (that is, springs) in the driest scenario and delay the switching on of cross-rift routes as climate ameliorates.

Implementation of travel distance is based on a simple 'energy quota'. Each starting agent has an energy value of 150. If the cost is zero (that is, the terrain flat) then they can travel 150 units (or cells) which is equivalent to 150 km given that

cell size is 1 by 1 km. The terrain is rarely flat and each cell traversed has a cost related to the topography which consumes the agent's energy proportionally thereby reducing the total travel distance before the agent's death. Supplementary Fig. 9b shows the variation in network connectivity with different travel distances and also the intra-variability at any chosen distance (see also Supplementary Fig. 10c). The inter-run variability is small compared to that between different travel distances. Other parameters included in the model are 'maximum turn angle'. Each agent places its back to the resource-patch from which it is hatched and moves forward. If the turn angle is low then there is little deviation in agent motion to the left or right. As the turn angle increases there is greater capacity for lateral movement and in theory for an agent to return to its source. Experimentation revealed that an angle of 20° gave the best compromise in terms of exploration to the right and left versus directed forward motion. The results are not sensitive to turn angle set unless it exceeds about 60° when a significant number of agents begin to lose their way and return home.

Other model inputs subject to parameter choices include the cost-map and cost-scaling. The impact of different cost-maps, variations of slope and roughness is shown in Supplementary Figs 10–13. The simulation in Supplementary Fig. 10 illustrates the differences between slope and roughness as cost layers. Roughness provides a more uniformly variable cost layer that is impacted less by the macro-scale geomorphology of the rift and more by local ground conditions that would be experienced by an agent moving over the terrain. The conclusions reported here use surface roughness as the primary cost layer. Slope restricts mobility to a greater extent, providing a more conservative set of results with respect to agent mobility. It is important to note that by using roughness our results favour mobility over isolation. The overall geographical pattern of results reported here do, however, not change in broad terms (Supplementary Figs 6,7,11) if slope is used, although the number of successful journeys does decline and the cross-rift linkages in particular around the Ethiopian Highlands cease under the present day dry scenario (Supplementary Fig. 7). Similarly by varying the scaling of cost used impacts on agent mobility (Supplementary Fig. 12); the scaling that most approximates Naismith's Rule was used (Supplementary Fig. 9). The contribution of each hydrological component was also modelled separately and these results are shown in Supplementary Figs 13 and 14. At each tick or time-step the model hatches for 4,000 agents who then attempt a journey. The model was routinely stopped after 100 ticks or after 400,000 attempted journeys. The results are not sensitive to the run-time beyond 100 ticks $\pm 20$, but connectivity can be reduced if the model is stopped prematurely, although most connections are established by 50 ticks. In summary, the model parameters include: (1) maximum travel distance in 3 days without water; (2) cost-scaling of the landscape reducing travel distance as the terrain become rougher; (3) the maximum turn angle set for an agent which determines the degree of lateral exploration versus forward motion; and (4) the model run-time which can impact on the probability of a connection being found by an agent between two resource points. The conclusions presented here are robust across a range of parameter values.

Outputs from the PATH model were analysed in three ways: (1) visually maps of successful agent journey are exported from Netlogo as raster files and uploaded for visualization and analysis in ArcMap; (2) within ArcMap the PATH output can be converted into a binary land cover map (crossed or not crossed) for analysis in Fragstats[68] which produces a range of connectivity variables for habitat patches; and (3) in addition the PATH model provides a matrix of successful journeys between named habitat (water) points which can be exported and plotted within Matlab (Fig. 6). Supplementary Tables 2–6 provides output metrics for the main model runs and associated sensitivity analysis. Supplementary Fig. 15 reports the results of a Principle Components Analysis on the all the metrics derived from the different model runs across the four climate scenarios. The 95% confidence ellipses clearly show the statistically significant differences between the four climate scenarios modelled. These differences are reinforced in Supplementary Table 6, which reports the mean model runs and the 95% error margins.

**Data availability.** Locations of springs, Matlab model code for estimating spring persistence, and the NetLogo ABM are available at: https://doi.org/10.6084/m9.figshare.c.3721141 (ref. 69)- GIS shapefiles for the digitized hydrological features used in the analysis are available from the authors on request.

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

## Acknowledgements

M.O.C. was financially supported by the European Community's Seventh Framework Programme (FP7/2007–2013) under the grant agreement no. 299091; T.G. was supported by the NSERC funding; we are grateful to Liz Flanary for GIS support and Weronika Wodzinska for providing hominin site location data. M.R.B. was supported by the UK Natural Environment Research Council (NE/H004211/1).

## Author contributions

M.O.C./T.G.: conceived the paper and designed the hydrological approach; C.J.M./M.O.C./T.G.: spring mapping and linked GIS analysis; M.O.C./C.J.M.: hydrological modelling; S.C.R./M.R.B./A.C.N.: design and implementation of the agent-based modelling and associated GIS preparations; M.O.C.: drafted the paper with all authors providing data interpretation and comments.

## Additional information

**Competing interests:** The authors declare no competing financial interests.

