## [Peer Review File · Nature Communications]

Reviewers' Comments:

Reviewer #1 (Remarks to the Author)

The paper tackles the interesting issue of how groundwater discharge through springs, river baseflow and oases varies in response to climate cycles. It then relates this to hominin dispersal in the east African Rift. The research is modelling based using mainly data from widely available global datasets with additional data digitised from national topographic maps. The model findings are that spring flow is controlled by geology not climate, and spring flow can persist for millennia through climate cycles. Similar work was reported by Cuthbert and Ashley 2014 where a mixture of observational data and modelling were used to highlight the importance of groundwater. Kuper & Kröpelin (2006) also addressed some issue around human occupation, climate and groundwater in the Eastern Sahara. The issue is, however, under-researched and new insights are welcome.

I am a groundwater scientist and will limit my comments to the groundwater aspects. Also, since the document is not line-numbered my comments are general. More specific editing would need a line numbered article.

I have several concerns with the article:

1. The mapping of the springs which leads to the conclusions about springs not being related to climate is weak and does not use the hydrogeological studies of others in the area which find an abundance of springs in the Highland areas. There is a useful study by Calow 10.1111/j.1745-6584.2009.00558.x which shows the availability of springs in highland areas is much higher than lowland areas. I would therefore question the source data of springs being mapped mainly in the rift valley.

2. The study does not draw on current research and experience of groundwater response to climate change Taylor et al. 2013, or groundwater response to drought (e.g. Calow 1997). There is also a growing literature on peoples current response to drying sources (e.g. Tucker et al 2014 for Ethiopia, and responses during current and previous El Nino) which could provide much more insight.

3 The study does not use information on the hydrogeology of the East African Rift, but rather uses a global dataset which is very poorly constrained for Africa. There are several datasets and many papers describing the hydrogeology of the different countries in question. Spring flow in these areas is highly complex and controlled by layering and anisotropy in the mainly crystalline rocks of the upper catchments, rather than a simply model using hydraulic diffusivity and assuming isotropic flow.

4 There is no attempt to calibrate or validate any of the groundwater models against any existing behaviour. This would give the paper much more weight. Currently, the confidence of the findings is much reduced, and doesn't justify the ascertains made in the paper about the robustness of the results. Is the model set up in such a way that groundwater discharge is almost guaranteed through the climate cycle. Ie given the input to groundwater through recharge, groundwater discharge will occur.

Therefore, although the paper addresses an interesting issue I cannot support its publication in Nature Communications in its current form - there are questions over its database of springs, the representation of the hydrogeology environment within the model, and the weight put on the uncalibrated model results.

Reviewer #2 (Remarks to the Author)

The major claims of this paper are that one should consider fresh water springs and groundwater fed rivers when thinking about how hominins may have used the landscape, as perhaps climate change is not the real driving force behind evolutionary change in hominins. The authors note that it is not climate change, but resource availability that is important to hominins. This is, of course, true, but resource availability is affected by climate change and it could be seen as an ultimate driver if not a direct driver. However, the authors then show that the persistence of these hydro-refugia may not be associated with climate changes, and in fact persist best in arid climates. All of that is fairly awesome and definitely new and of interest to those in paleoanthropology.

After the introduction of the agent-based model, however, there are sentences that do not make sense to me - with respect to how isolated populations survived. For example, -- "Importantly, networks of hydro-refugia maintain some gene-flow while also acting as evolutionary bottlenecks. Attempted dispersals during such dry periods would have been high risk and likely to lead to death of entire populations and, perhaps lineages."

Networks of hydro-refugia do not maintain gene flow. Populations of hominins may maintain gene flow between these networks. So, how can the networks maintain gene flow between hominin populations, and act as an evolutionary bottleneck at the same time? An evolutionary bottleneck is a type of genetic drift and gene flow is actually not a part of that - in fact, a bottleneck would occur if a limited number of hominins in a population survived through a drought with a result limited genetic variation. So then the next sentence is talking about attempted dispersals. Why would hominins attempt to disperse in the middle of a drought if they had water as it was stated that these refugia are mostly present when it is arid? Yes, it likely would have wiped out the population. So where is the gene flow? There is likely something here - I just feel it needs to be re-written to be clearer.

Later, the authors explain how following the various possible routes at certain times would get hominins to west Africa - also an interesting interpretation. Then, back to the evolutionary scenarios and, "In fact, one could argue that the lack of phylogenetic diversity in the hominin lineage has been overlooked and the potential for frequent, widespread dispersal illustrated here might explain this." Is there a lack of phylogenetic diversity in hominins in the Plio-Pleistocene? In the Miocene? Are the authors saying that there IS a lack of diversity and everyone has missed it? Again, clarity about what the authors are trying to say is needed here.

Finally, the following sentence is problematic: "This work does not discount the possibility that genetic drift driven by adaptation to variability may have been key in, for example, the evolution of larger brains in some taxa, but the potential for genetic mixing remains a challenge for this hypothesis to overcome 4." Usually, adaptation can be driven by genetic drift - I am trying to think how the authors might mean that genetic drift is driven by anything - since it is a random process.

I appreciate the author's three alternative possibilities and their urge to test some of the geological ideas presented in the paper. The closing paragraph is great.

The discussion and references to adaptation, gene flow, and genetic drift just need to be thought through more carefully - and I am not saying the authors haven't thought about it - it just needs to be presented so that a wider audience can understand what they are trying to say.

Kaye Reed

Reviewer #3 (Remarks to the Author)

The concept of "hydro-refugia" is an important contribution to the debates concerning environmental forces in human evolution, which has been dominated by the search for climatic correlations with speciation events, technological milestones, and other trends recorded in the hominin fossil record. The viewpoint and the modelling are original and it would be good to introduce these to a broader scientific audience, including paleoanthropologists.

I agree with the statement in the abstract that "...hominins were not adapting directly to shifting climates, but rather to the costs and returns of exploiting available resources." Initially, upon reading the title and abstract, I was positively impressed by this new perspective as an opportunity to widen the debate about processes that shaped human evolution. However, after reading the manuscript and looking over the supplementary figures, I cannot support publication without a major overhaul in both the presentation and factual basis for the results and conclusions. This relates more to assertions regarding how hydro-refugia might have influenced human evolution than to the modeled frequency and persistence of springs, although I also have a number of questions regarding the validity of those data.

Regarding the data on modern springs:

The estimate of 450 present-day springs is admittedly conservative (as stated in Caption to Fig. 1), and in looking at both Fig. 1 and the supplementary figures, it is not clear to me what the error factor is on the actual frequency of springs. How confident are the authors that they have captured a representative sample, and what about the number and distribution of springs that may not be documented in remote areas outside of the well-studied EAR and adjacent highlands? (In one day, I recently saw 3 springs in an area not included in their map, on the Laikipia Plateau in Kenya.)

The authors state that more springs persist for longer in dryland areas, based on their data (Fig. S1), but offer no geological or topographic explanation for this counter-intuitive result. I question the credibility of this finding. It strikes me that this could relate to the fact that more springs are known in such areas because that is where they are the most important resource for humans. They do not provide details about the database(s) that are the source of information on spring distribution and flow characteristics. Also, there are large dryland areas on the maps in Figs. 1 and 3 and the SOM that have no springs shown outside the EAR; this seems unlikely. The authors should consider and explain why the same areas that are drylands also are more geologically and topographically conducive to spring activity than the wetter areas.

Even given that geology and topography have a strong impact on spring persistence through drought cycles, climate must also affect (if not control) the number and persistence of springs through recharging of ground water during times of high rainfall and lake levels. The links between climate cycles and the geology and topography should be explored and discussed.

Other issues of concern:

- 1) Lake Turkana is mentioned as a large, freshwater lake. In fact, it is NOT fresh, but alkaline.
- 2) It is well-known that the rift valley differentially preserves evidence for hominins (and other organisms); there are sites outside the rift that support the widespread distribution of hominins outside of this tectonic province. The statement on Lines 105-106 is naïve and indicates that the authors are not familiar with current literature on the geological context of hominin evolution.
- 3) There is passing mention of the presence of paleo-springs and their supposed "poor preservation potential." In studies of the geological context of human evolution in Africa, there is an emerging appreciation for springs, not necessarily as "refugia" but as foci for plant, animal, and hominin activity. With this new realization, more paleo spring deposits are being recognized; their preservation potential in fact may be relatively high in many rift contexts. This could be a way to develop actual evidence to test their hypotheses, but that potential is more-or-less dismissed by

the authors.

Broader Issues:

A major failing of this paper is the strict uniformitarian approach asserting that what is recorded and modeled based on modern springs and rainfall can be applied wholesale to the millions of years of human evolution. There have been many geological events associated with the EAR and surrounding areas of the sub-continent during this time as well as variable climate cycles. These potential differences between present and past are completely ignored in the paper, based on the rationalization that today represents the "dry end" of the paleo-climate spectrum. Also, there is considerable latitudinal variation in climate along the EAR as well as variation over time in the major climate patterns; it cannot be considered as one big climatic province!

Overstatements (examples):

Lines 120-123. I take strong issue with the statement: "Our discovery of greatly increased and more geographically widespread groundwater hydro-refugia in such conditions confirms, for the first time, what had only been speculated previously or demonstrated for an isolated site with respect to how isolated populations might survive such climate extremes. Importantly, networks of hydro-refugia maintain some gene-flow while also acting as evolutionary bottlenecks."

This is vastly overstated and inappropriate. The authors have not discovered actual evidence from the geological record associated with hominins, as is implied in this statement, but instead are using modern patterns of springs to infer past numbers and distribution of "hydro-refugia." This can be presented as a hypothesis but it CANNOT be used as "evidence" for what actually happened in the past.

The following statement is not supportable based on current knowledge of hominin evolution: Lines 129-132. "Dispersal occurs first between areas of the rift floor and large rift-flanking streams/ rivers allowing movement initially transverse to the rift axis, prior to movement along the main rift axis. This unexpected finding is at odds with the most common assumption of along-rift dispersal but shows agreement with westward dispersals observed in some hominins and genetic studies of several other species." We have NO idea about hominin dispersals based on the evidence in the literature cited, which is simply a hominin fossil occurrence outside of the rift system. There is no way to infer whether the species dispersed from somewhere else or was endemic to this area. We do not know where in Africa hominins originated. This statement reveals that the authors know relatively little about what is and is not known about human evolution; their statements and proposals are based on superficial knowledge of the literature.

Also, many other statements in the paper are presented as "facts," e.g., "...as the climate becomes wetter, trans-rift dispersal routes become active prior to those along the rift-axis." This may be meant as a proposal based on the model, but throughout the text such statements are presented as if they are fact; what should be proposed as hypothesis becomes assertion. This would be acceptable if presented consistently as conditional based on the proposed model, but it is not acceptable presented as fact.

There are many ways that tectonics, topography, and climate cycles could affect resources important to hominins (and other organisms). The importance of springs/hydro-resources does not need to be set in opposition to climate as a controlling factor in human evolution. Rather, this paper should be re-written to present the hydro-geological models and resource utilization network maps as interesting "food for thought" that can add to the debates about forces that shaped hominin evolution.

I have also included sticky-note comments in the attached pdf.

Reviewer #4 (Remarks to the Author)

The paper examines the role of groundwater hydro-refugia in hominin dispersal by: 1) reconstructing -via hydrological analysis- the distribution of hydro-refugia under different climatic conditions; and 2) modelling the connectivity/isolation of hominin groups via Agent-Based simulation. The main, intriguing conclusion suggested by the authors is that shifts between arid and wet climate conditions translated into different degree of habitat connectivity. I think this is an interesting paper with some potentially relevant implications on human metapopulation structure and evolution.

Given that hydrological analyses is not within my domain of expertise, I will focus my review on aspects pertaining the computational model proposed. The authors used a published agent-based model (Pathway Analysis Through Habitat, PATH), designed to detect patterns of successful inter-habitat migration. The key assumptions of PATH is that agents move stochastically between patches but have limited mobility range (dictated by environmental and biological constraints). Here patches are represented by the hydro-refugia detected by the hydrological analysis, the environmental constraints by the topography, and the biological constraint by presumed physiological limits (e.g. number of days of travel without water). Based on the simulation output the authors suggest an alternation between periods of high connectivity (i.e. most patches connected by migration routes) and periods with lower connectivity when the dispersal of agents are limited to local clusters of hydro-refugia.

While I think that the overall conclusion of the model is relevant and robust, I do have a number of comments, which I summarise in four points

1. Is an ABM really necessary in this case?

The key objective of PATH is to determine where we should expect to observe migration routes, definitely a better alternative to traditional GIS-based least-cost pathway (that offers only connectors rather than corridors) but somewhat very similar to other more widely used methods based on circuit theory (e.g. McRae et al 2008, *Ecology* 10: 2712-2724.). From what I understand here, the primary objective is here to determine whether the hydro-refugia are connected or disconnected into multiple isolated clusters. If that is the case, I wonder why alternative (more common) methods were not considered. For example, one could generate a matrix defining the time required to reach from a patch *i* any other patch *j*, using the topographic constraints presented here, then determine which pairs can be "linked" given the physiological assumptions. The resulting matrix can then be used to create a simple graph that should be comparable to the output of PATH. The only difference I see is that PATH can generate a network with weighted edges recording the number of successful in migration and out migration from each patch. Perhaps I am missing something, but I would like to know why this specific approach has been used to measure patch connectivity, given its computationally less efficient, it's subject to stochastic components that are not explored, and have a higher number of parameters.

2. Lack of quantitative measures of connectedness

The authors explore the simulation output purely in visual terms. While the results shown on figure 3 is undoubtedly compelling, it would help to quantify the level of patch connectivity in numerical terms. For connected graphs this could be something relatively simple (e.g. network diameter, node connectivity, link connectivity) while for disconnected graph even reporting the number of disconnected subgraphs might be helpful. Alternatively, one could benefit from the PATH model and record dispersal success, search time, or cell immigration (e.g. Tischendorf and Fahrig 2000, *Landscape Ecology*, 15, 633-641). In any case, there seems to be a wide range of options (see Kindlmann and Burel 2008, *Landscape Ecology*, 23, 879-890 for review) and I think a quantifiable description would allow a more formal parameter exploration and sensitivity analysis.

3. Modelling pedestrian movement

The authors create a cost layer using Naismith's Rule, which converts terrain properties into walking speed. Thus the same "as the crow flies" distance would take a different amount of time depending on the terrain properties. But then they refer to their model parameters in km (line 325) rather than in hours. The authors are bit vague on this but I understand their assumption is a maximum of three days (line 314 "a maximum travel time of 3 days without water), with 10 hours spent on walking each (lines 321-322 "[...] assume that daily walking distance [...] over a 10-hour period). Now suppose we have two patches with an inter-distance of 120km but with a very rough terrain in the middle that requires more than 30 hours of travel time. Would the agent manage to reach its destination? I might have missed something, but the authors need to clarify how this aspect of the model has been implemented. Are they using a 30-hours limit or are they using specific distances (e.g. 150km) but ignoring the amount of time taken, or are they using the smallest between the two? Without the source code (see next section) it is hard to understand how this was implemented. A second minor remark is the assumption of a 30 hours walking time without water. I wonder whether it is right to assume a constant performance in physical activity without any water intake. I think a better way to approach this (and overcome possible issues/critiques with the model assumptions) is to identify the range of conditions (parameter settings) that can generate a specific outcome (e.g. a certain degree of patch connectivity). An example of this kind of approach is used by Gallagher et al 2015, PNAS, 112, 14218-14223.

4. Model Source Code

I could not find the original code of PATH developed by Hargrove and Westervelt, nor did the authors provided their implementation. There are certainly parameter settings (e.g. the number of agents, the stopping time, etc.) that are not crucial but nonetheless affect the outcome of the simulation. In order to make their work reproducible (and clarify any issues such as the one mentioned above on the pedestrian movements) I think it is paramount that the source code is shared, along with the specifics of the experiment design on an online repository (e.g. github, openABM) or in the Supplementary Information.

(Reviewers' comments in black, author responses in red)

Reviewer #1 (Remarks to the Author):

The paper tackles the interesting issue of how groundwater discharge through springs, river baseflow and oases varies in response to climate cycles. It then relates this to hominin dispersal in the east African Rift. The research is modelling based using mainly data from widely available global datasets with additional data digitised from national topographic maps. The model findings are that spring flow is controlled by geology not climate, and spring flow can persist for millennia through climate cycles. Similar work was reported by Cuthbert and Ashley 2014 where a mixture of observational data and modelling were used to highlight the importance of groundwater. Kuper & Kröpelin (2006) also addressed some issue around human occupation, climate and groundwater in the Eastern Sahara. The issue is, however, under-researched and new insights are welcome.

Thank you to the reviewer for acknowledging the significant contribution the paper can make to the literature on this subject.

I am a groundwater scientist and will limit my comments to the groundwater aspects. Also, since the document is not line-numbered my comments are general. More specific editing would need a line numbered article.

I have several concerns with the article:

1. The mapping of the springs which leads to the conclusions about springs not being related to climate is weak and does not use the hydrogeological studies of others in the area which find an abundance of springs in the Highland areas. There is a useful study by Calow 10.1111/j.1745-6584.2009.00558.x which shows the availability of springs in highland areas is much higher than lowland areas. I would therefore question the source data of springs being mapped mainly in the rift valley.

The reviewer is correct that scale of the mapping necessarily used for this regional study inevitably misses springs which may have been apparent from more local maps. However, since smaller scale maps are not consistently available across the region, we opted for the use of larger scale maps at the most consistent possible scales in order to minimise any potential mapping bias. We have done some comparisons of springs from overlapping maps at different scales to determine the representativeness at the scales used - see reply to Reviewer 3 below for details. We have also demonstrated that our sample is statistically representative of the range of groundwater response times, which is important for our arguments about spring persistence.

With regard to the specific concern about springs being mapped mainly in the rift valley and not highland areas, we note that many springs mapped in Ethiopia (using the same map series for the whole country) are located away from the main rift valley axis (Figure 1). It is possible that there may be potential bias in the mapping of springs towards drier areas stemming from the relative importance of springs in drier areas as opposed to areas where water is more easily accessible. We consider this possibility to be compounded by the presence of springs associated with stream channels which flow persistently in areas such as the Ethiopian highlands which the reviewer mentions. There is caveat in this regard in the caption to Fig 1.

In terms of how this relates to the conclusions about the relative importance of climate – we would like to clarify that the paper does not conclude that ‘springs are not related to climate’. Rather, our intention is to show that climate is not the primary factor in spring persistence but that geology and topography are just as, if not more, important controls. It is of course true that some groundwater recharge is a necessary condition for a spring to exist and so the climate must produce enough precipitation so that not all of this input is completely lost to runoff or evapotranspiration. However, this is not always a sufficient condition for producing a persistent spring. Rather, hydrologic theory clearly shows that it is the groundwater response time which governs the responsiveness of groundwater discharge for a given climate forcing and thus the persistence in time. So, for example, in wetter areas with greater recharge, shallow water tables may lead to a greater density of ‘gaining’ streams (i.e. receiving groundwater discharge) and thus shorter flow paths and smaller groundwater response times. This is exactly what is shown by the useful Calow *et al.* paper that the reviewer mentions (and which we now cite): in that context the aquifer flow paths are often small, thus, despite the recharge being higher, many of the reported springs dry up during extended drought periods with those at the lower topographic elevations and longer flow paths being more reliable sources during drought.

Thus we maintain that the data are robust with regard to demonstrating a lower importance for climate in terms of control on water availability during dry periods than is currently assumed - the geological and topographical characteristics are also fundamental controls. This is clearly shown by Figure S1. If wetter areas substantially produced more springs, and/or more persistent, springs, there would be an observable deviation in the distribution away from the distribution of the ‘regional recharge rate’. In fact, there is a slight bias in the opposite direction.

To make these points clearer we have added more explanation and clarity regarding the various controls on spring flow in the main text. We have also made it clearer that we are not discounting climate as an important factor in spring presence and persistence but rather that the hydrogeological conditions (geology and topography) may be at least as important. Finally we have added more detail on the representativeness of the mapping in the methods and SI.

2. The study does not draw on current research and experience of groundwater response to climate change Taylor *et al.* 2013, or groundwater response to drought (e.g. Calow 1997). There is also a growing literature on peoples current response to drying sources (e.g. Tucker *et al.* 2014 for Ethiopia, and responses during current and previous El Nino) which could provide much more insight.

Thank you for highlighting this area of research - the authors are very aware of its importance having published work in this field ourselves. We agree that it is useful to incorporate recent work on the groundwater response to climate change such as Taylor et al. 2013 and have done so in the text. However, we are hesitant to make direct comparisons between the modern experience and the paleoecological context of our hominin ancestors for several reasons. For example, there would have been major differences in the population density and community structure of hominins and the technology available for collecting and transporting water - hominins being limited to accessing natural discharges at the site only and modern humans being able to withdraw from wells and boreholes as well as store and carry. The timescales of change we are considering here, over precessional cycles, are also much longer than we can observe in the modern context. So, while we conclude that the hydrological framework is a useful context in which to consider modern day responses to climate change, we think it would be dangerous to assume the response of modern people is analogous to that of hominins.

3 The study does not use information on the hydrogeology of the East African Rift, but rather uses a global dataset which is very poorly constrained for Africa. There are several datasets and many papers describing the hydrogeology of the different countries in question. Spring flow in these areas is highly complex and controlled by layering and anisotropy in the mainly crystalline rocks of the upper catchments, rather than a simply model using hydraulic diffusivity and assuming isotropic flow.

With respect to the first criticism regarding the data used, Macdonald et al. (MacDonald, A M, Bonsor, H C, Ó Dochartaigh, B É, and Taylor, R G. 2012. Quantitative maps of groundwater resources in Africa. *Environmental Research Letters*, 7 (2), 024009) is the seminal and valuable resource that compiled available African hydrogeological datasets. This has also been recently been extended in the form of the Africa Groundwater Atlas (<https://www.bgs.ac.uk/africagroundwateratlas/>). We returned to these resources to re-check that we were not missing any more detailed African-centric hydrogeology data applicable to our study. While being a very important repository of local study information (papers and reports), its piecemeal nature gives a small spatial coverage regionally, and generally does not quantify the variables that are important for our model (i.e. permeability and porosity). The resource we have used for mapping hydraulic parameters (Gleeson, T., Moosdorf, N., Hartmann, J. & Beek, L. A glimpse beneath earth's surface: GLobal HYdrogeology MaPS (GLHYMPS) of permeability and porosity. *Geophys Res Lett* 41, 3891-3898, 2014) used identical available geological base mapping as Macdonald et al. (2012) (i.e. Persits, Feliks M., et al. Maps showing geology, oil and gas fields and geological provinces of Africa. No. 97-470-A. Geological Survey (US), 1997) but has the added advantage of allowing quantification of the parameters needed for the modelling on a regionally consistent basis. Our mapping also uses the same recharge rates as used in Macdonald et al. (2012).

With respect to the second criticism regarding the use of an isotropic model we agree that local features such as layering and fracturing will influence spring flows. However, the use of a homogeneous model is appropriate and robust for our purposes in this paper for several

reasons. First, Swanson et al. (2004) (Swanson, Susan K., and Jean M. Bahr. "Analytical and numerical models to explain steady rates of spring flow." *Ground Water* 42.5 (2004): 747-759) carried out a direct comparison of an analytical spring flow model, with identical underlying assumptions as our model, with a more complex numerical model which included explicit representation of high permeability layering. They conclude that the inclusion of such complexity does not affect the bulk behaviour of the aquifer with respect to the transience in the spring discharge, since their explicit representation does not increase the effective transmissivity for use in the analytical solution enough to cause significant changes in the overall groundwater response time. Given this fact, and the fact that the hydraulic parameters have been mapped on a regional scale, the use of more detailed model structure with additional parameter requirements at a higher spatial resolution would not be defensible in our view.

We have added additional information on the mapping and modelling in light of the above points to strengthen the choice of our approach in the methods section.

4 There is no attempt to calibrate or validate any of the groundwater models against any existing behaviour. This would give the paper much more weight. Currently, the confidence of the findings is much reduced, and doesn't justify the ascertains made in the paper about the robustness of the results.

We appreciate the reviewer's concern here and have therefore carried out significant additional analyses to test the models used. Despite consulting a number of leading international authorities on springs such as Abe Springer, and the 'Spring Stewardship' organisation who administrate the largest global spring database (<http://springsdata.org/>) we have found very little published data on spring flows in the region to rigorously validate the models even for short timescales. Over the longer timescales of precessional climate cycles relevant to this paper, of course no direct records exist. However we have found several ways in which the model can be tested and have added a new 'Model testing' section to the methods along with a new supplementary figure describing this work.

In summary defence of our modelling approach, we have now demonstrated that the model has a sound theoretical basis and the parsimonious model structure is consistent with the use of parameters derived from regional mapping. The models are uncalibrated and we thus avoid the problem of being overfitted to a limited number of data points and then wildly extrapolated. The model approach performs well in tests against the limited hydrological data, model output and geological evidence that is available in the region for constraining spring flow persistence. We therefore consider the use of our model for constraining order of magnitude persistence of springs to be appropriate and robust.

Is the model set up in such a way that groundwater discharge is almost guaranteed through the climate cycle. Ie given the input to groundwater through recharge, groundwater discharge will occur.

The reviewer is correct that some springs have large enough catchments so that even a small amount of recharge during the driest periods will keep the spring flowing above the threshold we have defined for a 'productive' spring. However, a smaller catchment with a lower average flow rate might also keep flowing above the same threshold if its groundwater response time is sufficiently long to buffer the climate variability. Thus in most cases, the degree of persistence during the driest periods is a complex interplay between the input recharge, the catchment area and the hydraulic properties of the spring in question. The models we have applied are powerful in allowing an integration of all these factors to enable the distribution of the timescales of persistence to be derived. Statistically we find that groundwater response time is a better predictor of spring persistence than the prevailing climate (Table S1).

Therefore, although the paper addresses an interesting issue I cannot support its publication in Nature Communications in its current form - there are questions over its database of springs, the representation of the hydrogeology environment within the model, and the weight put on the uncalibrated model results.

We hope our responses and manuscript changes have now adequately addressed the issues which the review raises – i.e. that we have used appropriate data and models suited to our purpose in this paper in deriving a sound conceptual framework for typical configurations of water availability for wet to dry periods.

Reviewer #2 (Remarks to the Author):

The major claims of this paper are that one should consider fresh water springs and groundwater fed rivers when thinking about how hominins may have used the landscape, as perhaps climate change is not the real driving force behind evolutionary change in hominins. The authors note that it is not climate change, but resource availability that is important to hominins. This is, of course, true, but resource availability is affected by climate change and it could be seen as an ultimate driver if not a direct driver. However, the authors then show that the persistence of these hydro-refugia may not be associated with climate changes, and in fact persist best in arid climates. All of that is fairly awesome and definitely new and of interest to those in paleoanthropology.

Many thanks to the reviewer for recognising the significance of our findings in the paper.

After the introduction of the agent-based model, however, there are sentences that do not make sense to me - with respect to how isolated populations survived. For example, -- "Importantly, networks of hydro-refugia maintain some gene-flow while also acting as evolutionary bottlenecks. Attempted dispersals during such dry periods would have been high risk and likely to lead to death of entire populations and, perhaps lineages." Networks of hydro-refugia do not maintain gene flow. Populations of hominins may maintain gene flow between these networks. So, how can the networks maintain gene flow between hominin populations, and act as an evolutionary bottleneck at the same time?

We are grateful for the reviewer's challenge to bring greater clarity on these points and better focus on the core message of the paper. These sentences have been rewritten in the text with respect to a new conceptual Figure 7 to reflect the reviewer's points, as described in our responses which follow.

An evolutionary bottleneck is a type of genetic drift and gene flow is actually not a part of that - in fact, a bottleneck would occur if a limited number of hominins in a population survived through a drought with a result limited genetic variation. So then the next sentence is talking about attempted dispersals. Why would hominins attempt to disperse in the middle of a drought if they had water as it was stated that these refugia are mostly present when it is arid? Yes, it likely would have wiped out the population. So where is the gene flow?

We have simplified our discussion of the evolutionary consequences in the paper to more clearly emphasise the existence of hydrorefugia that these regions offer in times of aridity. We have omitted the mention of potential evolutionary bottlenecks and failed dispersals so that the central message of the paper is not muddled, and remains on the beneficial effects of these regions in times of aridity in supporting hominin populations.

There is likely something here - I just feel it needs to be re-written to be clearer. Later, the authors explain how following the various possible routes at certain times would get hominins to west Africa - also an interesting interpretation. Then, back to the evolutionary scenarios and, "In fact, one could argue that the lack of phylogenetic diversity in the hominin lineage has been overlooked and the potential for frequent, widespread dispersal illustrated here might explain this." Is there a lack of phylogenetic diversity in hominins in the Plio-Pleistocene? In the Miocene? Are the authors saying that there IS a lack of diversity and everyone has missed it? Again, clarity about what the authors are trying to say is needed here.

We have rewritten this sentence and cited the relevant papers accordingly. There is debate about whether or not the levels of phylogenetic diversity is too high or not. However, that is not the central point of our study, so we have chosen to focus on the fact that the hydro-refugia model offers a means 1) to understand the survival of hominin populations in arid periods and 2) to point out that these patterns may be relevant to how the diversity arose, not necessarily how much or how little diversity there is.

Finally, the following sentence is problematic: "This work does not discount the possibility that genetic drift driven by adaptation to variability may have been key in, for example, the evolution of larger brains in some taxa, but the potential for genetic mixing remains a challenge for this hypothesis to overcome 4."

We have removed this sentence.

Usually, adaptation can be driven by genetic drift - I am trying to think how the authors might mean that genetic drift is driven by anything - since it is a random process. I appreciate the author's three alternative possibilities and their urge to test some of the geological ideas

presented in the paper. The closing paragraph is great. The discussion and references to adaptation, gene flow, and genetic drift just need to be thought through more carefully - and I am not saying the authors haven't thought about it - it just needs to be presented so that a wider audience can understand what they are trying to say.

This has been done, and the text reflects more clearly what we think the central importance of the hydro-refugia model is: i.e. facilitating the survival of hominins during arid periods.

Reviewer #3 (Remarks to the Author):

The concept of "hydro-refugia" is an important contribution to the debates concerning environmental forces in human evolution, which has been dominated by the search for climatic correlations with speciation events, technological milestones, and other trends recorded in the hominin fossil record. The viewpoint and the modelling are original and it would be good to introduce these to a broader scientific audience, including paleoanthropologists.

I agree with the statement in the abstract that "...hominins were not adapting directly to shifting climates, but rather to the costs and returns of exploiting available resources." Initially, upon reading the title and abstract, I was positively impressed by this new perspective as an opportunity to widen the debate about processes that shaped human evolution. However, after reading the manuscript and looking over the supplementary figures, I cannot support publication without a major overhaul in both the presentation and factual basis for the results and conclusions. This relates more to assertions regarding how hydro-refugia might have influenced human evolution than to the modeled frequency and persistence of springs, although I also have a number of questions regarding the validity of those data.

Many thanks to the reviewer for the encouragement to improve our paper to meet the challenges raised.

Regarding the data on modern springs:

The estimate of 450 present-day springs is admittedly conservative (as stated in Caption to Fig. 1), and in looking at both Fig. 1 and the supplementary figures, it is not clear to me what the error factor is on the actual frequency of springs. How confident are the authors that they have captured a representative sample, and what about the number and distribution of springs that may not be documented in remote areas outside of the well-studied EAR and adjacent highlands? (In one day, I recently saw 3 springs in an area not included in their map, on the Laikipia Plateau in Kenya.)

As stated in our responses to Reviewer 1, we acknowledge that there may be possible biases in mapping and have added appropriate caveats to the text. Furthermore, we have carried out tests on areas mapped at different scales to look at the bias introduced by using different map scales as well as statistical tests to assess how representative our sample of springs is in determining the correct frequency distribution of spring persistence.

First, we have directly compared the number of springs mapped at 1:125 000 and 1:500 000 for the same geographical areas where both sets of maps could be found. Where this was possible in Northern Tanzania the average underestimate in the number of springs at the 1:500 000 scale was 60% in comparison to the 1:125 000 maps. Experience of the authors in cross checking the locations of springs from the 1:125 000 maps against local Masai knowledge, suggests they are remarkably accurate across several map sheets in representing the main sources of water used by local people. However, the scale of mapping possible across the modelled area generally decreased in resolution from 1:125 000 in Tanzania, to 1:250 000 in Kenya, and 1:500 000 in Ethiopia. Thus, we expect that the lower number of springs mapped due to the changing map scales leads to a loss of accuracy in the absolute spring count from nearly 100% accuracy in the south of the area to around 40% in the north.

Second, we have randomly resampled the total set of springs for increasing subsample size as a proportion of the total spring set. For each subsample we have calculated the RMSE for the cumulative frequency distribution of groundwater response times (GRT – as shown in Figure 4) of the subsample against the full spring set. We then plotted the RMSE as a function of the proportion of the total sample size. The result is now given in Figure S4 which indicates that the spring sample becomes representative at around 75% of the full sample size. This demonstrates that the sample of springs we have is representative of range of groundwater response times and thus of spring persistence across the study area.

In summary, we have been able to quantify the likely underestimation of the absolute number of springs which increases from south to north in the modelled area. This is consistent with the observations by Reviewers 1 and 3 of ‘missing’ springs known to them. However, of more importance to the paper is that the sample taken is statistically representative of the spring persistence. Thus, while the exact spatial patterns of springs may not be correct, which is inevitably the case anyway given the uniformitarian assumptions we make for the ABM modelling, the conceptual framework for typical configurations of water availability for wet to dry periods is robust.

The authors state that more springs persist for longer in dryland areas, based on their data (Fig. S1), but offer no geological or topographic explanation for this counter-intuitive result. I question the credibility of this finding. It strikes me that this could relate to the fact that more springs are known in such areas because that is where they are the most important resource for humans.

We have now added more explanation of this result in the main text. See comments above and in response to Reviewer 1 with respect to mapping bias.

They do not provide details about the database(s) that are the source of information on spring distribution and flow characteristics.

We are unsure of the reviewer's concern here. It was because there were no existing databases of springs for the region that we undertook the spring mapping and parameterisation described in the methods section. Since we have generated an important database for an otherwise data-sparse region, we will make our data publically available via <http://springsdata.org/> should the paper be accepted for publication.

Also, there are large dryland areas on the maps in Figs. 1 and 3 and the SOM that have no springs shown outside the EAR; this seems unlikely. The authors should consider and explain why the same areas that are drylands also are more geologically and topographically conducive to spring activity than the wetter areas.

It is to be expected that large areas have no springs – and more so in drylands where water tables tend to be deep and thus places where the water table intersects the topography to enable groundwater discharge are more widely spaced. Figure 2 shows there is no strong preference for springs to be in any particular climate zone – the distribution of springs approximates the distribution of climate, which is one line of evidence that climate is not the primary control on spring distribution.

Even given that geology and topography have a strong impact on spring persistence through drought cycles, climate must also affect (if not control) the number and persistence of springs through recharging of ground water during times of high rainfall and lake levels. The links between climate cycles and the geology and topography should be explored and discussed.

We have now added a longer explanation to the main text of the controls on spring flow with regard to the relative importance of climate, topography and geology.

Other issues of concern:

1) Lake Turkana is mentioned as a large, freshwater lake. In fact, it is NOT fresh, but alkaline.

The reviewer is correct that Lake Turkana is alkaline, but it is also relatively fresh in comparison with many other rift valley alkaline lakes which tend to be hyper-saline. Lake Turkana's salinity is around 2500 ppm which makes it 'fresh' by some measures (e.g. it supports freshwater fish populations) and 'brackish' by others, but not saline. For the purposes of this paper, we consider it to be a potable resource with respect to hominins and have now made this assumption explicit in the text.

2) It is well-known that the rift valley differentially preserves evidence for hominins (and other organisms); there are sites outside the rift that support the widespread distribution of hominins outside of this tectonic province. The statement on Lines 105-106 is naïve and indicates that the authors are not familiar with current literature on the geological context of hominin evolution.

The authors are aware of the literature with regard to preservation bias (as well as exposure and prospection bias) for hominins in the rift valley. The statement in the text the reviewer refers

to was intended in support of this fact, not claiming to be saying something new, which indeed would be naive. However, since it is not the main focus of the paper, this statement has now been deleted.

3) There is passing mention of the presence of paleo-springs and their supposed "poor preservation potential." In studies of the geological context of human evolution in Africa, there is an emerging appreciation for springs, not necessarily as "refugia" but as foci for plant, animal, and hominin activity. With this new realization, more paleo spring deposits are being recognized; their preservation potential in fact may be relatively high in many rift contexts. This could be a way to develop actual evidence to test their hypotheses, but that potential is more-or-less dismissed by the authors.

We are not entirely sure what the reviewer is driving at here. Some of the authors have been trying to make this 'realisation' more widespread for many years, and are not trying to dismiss the idea at all. We consider springs to have a relatively low preservation potential since groundwater itself leaves no geological trace unless conditions are just right for precipitation of minerals which record a freshwater signature. Regularly visiting modern springs in the rift valley and beyond, the authors note that the precipitation of minerals occurs in a relatively small percentage of such environments. Thus in the geological record they are difficult to distinguish from wetlands of other types and their associated sedimentation/flora/faunal assemblages. Just because preservation may be low, we agree that this evidence should not be overlooked where it does exist, and we have therefore modified the text to make a more positive statement in this regard.

Broader Issues:

A major failing of this paper is the strict uniformitarian approach asserting that what is recorded and modeled based on modern springs and rainfall can be applied wholesale to the millions of years of human evolution. There have been many geological events associated with the EAR and surrounding areas of the sub-continent during this time as well as variable climate cycles. These potential differences between present and past are completely ignored in the paper, based on the rationalization that today represents the "dry end" of the paleo-climate spectrum. Also, there is considerable latitudinal variation in climate along the EAR as well as variation over time in the major climate patterns; it cannot be considered as one big climatic province!

We agree that the EARS cannot be treated as one big climatic province and that geological and climatic change has of course occurred over the last few million years. Since it is not currently possible to accurately reconstruct the spatial-temporal variability of the paleoclimate or the paleo-landscape over millions of years, we have rather used the current landscape as a laboratory to understand the controls on the potential for hominin dispersal and isolation. This is indeed an application of the fundamental geological principal of uniformitarianism as we have now made explicit in the text: the specific location and geometry of landscape elements (i.e. lakes, rivers and springs) will have changed through time, but the basic elements can be shown from the geological/environmental record to have always been present. Thus, the conceptual

framework we present (see new Figure 7) is not an attempt to model any particular time & place but is a novel and powerful basis for future work to use as a basis for exploring specific contexts as more field data become available.

Overstatements (examples):

Lines 120-123. I take strong issue with the statement: "Our discovery of greatly increased and more geographically widespread groundwater hydro-refugia in such conditions confirms, for the first time, what had only been speculated previously or demonstrated for an isolated site with respect to how isolated populations might survive such climate extremes. Importantly, networks of hydro-refugia maintain some gene-flow while also acting as evolutionary bottlenecks."

This is vastly overstated and inappropriate. The authors have not discovered actual evidence from the geological record associated with hominins, as is implied in this statement, but instead are using modern patterns of springs to infer past numbers and distribution of "hydro-refugia." This can be presented as a hypothesis but it CANNOT be used as "evidence" for what actually happened in the past.

We have moderated the language as the reviewer suggests.

The following statement is not supportable based on current knowledge of hominin evolution: Lines 129-132. "Dispersal occurs first between areas of the rift floor and large rift-flanking streams/rivers allowing movement initially transverse to the rift axis, prior to movement along the main rift axis. This unexpected finding is at odds with the most common assumption of along-rift dispersal but shows agreement with westward dispersals observed in some hominins and genetic studies of several other species." We have NO idea about hominin dispersals based on the evidence in the literature cited, which is simply a hominin fossil occurrence outside of the rift system. There is no way to infer whether the species dispersed from somewhere else or was endemic to this area. We do not know where in Africa hominins originated. This statement reveals that the authors know relatively little about what is and is not known about human evolution; their statements and proposals are based on superficial knowledge of the literature.

While, due to the choice of double-blind peer review, we can't demonstrate the authors' track record in this field including a deep knowledge of the literature, we appreciate the reviewer's caution here – indeed there are many unknowns and there is very little evidence with which to frame this debate. However, our contribution here is to offer new insights into how this question might be explored further, in the absence of new geological evidence, or in combination as new evidence (hopefully) emerges. A better understanding of how critical resources such as water may have been available in the landscape is a new approach which we believe can be further developed and explored to test paleoecological models of hominin evolution and dispersal. We are offering a novel environmental framework and models with which to overcome the impasse due to the lack of hominin finds outside the rift valley.

Also, many other statements in the paper are presented as "facts," e.g., "...as the climate becomes wetter, trans-rift dispersal routes become active prior to those along the rift-axis." This may be meant as a proposal based on the model, but throughout the text such statements are presented as if they are fact; what should be proposed as hypothesis becomes assertion. This would be acceptable if presented consistently as conditional based on the proposed model, but it is not acceptable presented as fact.

We have added the word 'modelled' to deal with this in several places in the text to make clear these are model outcomes not 'facts' as the reviewer understands them.

There are many ways that tectonics, topography, and climate cycles could affect resources important to hominins (and other organisms). The importance of springs/hydro-resources does not need to be set in opposition to climate as a controlling factor in human evolution. Rather, this paper should be re-written to present the hydro-geological models and resource utilization network maps as interesting "food for thought" that can add to the debates about forces that shaped hominin evolution.

We have moderated the text, to avoid setting up the importance of springs/hydro-resources in opposition to the 'climate-forcing' hypothesis but rather that the insights of our mapping and modelling provide a new framework for understanding what role climate variability might have in controlling water resources during periods of aridity. As such, we have changed a number of key phrases in the text to make this clearer.

I have also included sticky-note comments in the attached pdf.

Copies of the reviewer's sticky-note comments by line number have been pasted below and our response added.

Line 40 - climate-forcing hypothesis, as stated above. There is no necessary dichotomy between these; both could have been operating

As above, we agree and have made this clearer.

Line 50 - potential correlates of

We have amended the text as suggested.

Line 60 - Average size and range of these areas?

We haven't defined the exact areas here since we describe the aridity distribution later in the text with respect to Figure 2.

Line 64 - Lake Turkana is NOT fresh - I am surprised and a bit dismayed that the authors do not know it is alkaline.

See our response to the same point (numbered point 1) above.

Line 75 - But then why should such geological conditions be biased toward "dryland" areas?

We have now added more explanation of this in the text.

Line 94 - This is too strong - the present doesn't necessarily predict the past this directly, certainly not over the past 6 million years of hominin evolution. The results can suggest this, but do not demonstrate it.

We have replaced 'demonstrate' with 'suggest'

Line 96 - potable, or fresh water lakes.

We have amended the text as suggested.

Line 106 - It is not a matter of "may." The hominin fossil record is concentrated in the rift valley for well-known and accepted geological reasons. The authors reveal their lack of exposure to paleontological and paleanthropological literature in this rather naive statement.

See our response above to the same point.

Line 110 - Appropriate caution here.

Ok.

Line 117 - But there is huge latitudinal variation in this sub-continental region; it is not one climatic entity!

Indeed. See our response above to the same point.

Line 119 - No, you have not discovered evidence from the actual geological record associated with hominins, but rather are using modern patterns of springs to infer the past numbers of "hydro-refugia." This claim is overstated.

See our response above to the same point.

Line 120 - Yes, it appears that there would be a lot of springs during dry conditions comparable to those of today, and this is interesting. But it still makes a big uniformitarian assumption that should be acknowledged. Aquifers, groundwater recharge rates, hydrothermal spring activity, etc., as well as the spatial distribution of springs, all can be affected by tectonics and climate, which are quite variable over geological time.

See our response above to the same point.

Line 122 - based on our model,

This sentence has been removed.

Line 123 – would - These statements must be qualified - they are not facts but predictions and inferences using modern evidence.

This sentence has been removed.

Line 128 - Again, must be qualified. "We propose that dispersal would occur...." Also, this assumes that water drives everything. It is important, but there are many other factors on the landscape that would affect hominin dispersals and habitat use.

We have amended the text as suggested.

Line 129 - perhaps promoting...

We have amended the text as suggested.

Line 132 - We have NO idea about hominin dispersals based on the evidence in this cited literature. This is simply a hominin fossil occurrence outside of the rift system; there is no way to infer whether the species dispersed from somewhere else or was endemic to this area. We do not know where in Africa hominins originated.

See our response above to the same point.

Line 133 – modern

We have amended the text as suggested and have also inserted the word 'modelled' into this sentence to be clear this is an inference from the model.

Line 139 - Indeed! Yet a lot of the previous argument is based on the assumption that the first thing a hominin population would do would be to follow new water pathways. This is not necessarily true.

Yes, that is why this caveat is there in the text. We have added here additional caveats regarding the uniformitarian issue as discussed above.

Line 149 - Really? Some would say that there was a lot of diversity, i.e., we now know that the phylogenetic pattern is more of a bush than a simple tree. More hominin taxa are being added every year. On the other hand, proposing dispersal patterns over time that could relate to hydro-refugia is a positive contribution to the debate.

Please see our responses to reviewer 2 on this same point.

Line 161 - when faced by...

We have amended the text as suggested.

Line 162 – could

We have amended the text as suggested.

Line 464 - Very important to note that the number of springs is, and likely was, much greater than shown on the map.

Yes, we agree and we have added this to the methods section now rather than this just appearing in the figure legend.

Reviewer #4 (Remarks to the Author):

The paper examines the role of groundwater hydro-refugia in hominin dispersal by: 1) reconstructing -via hydrological analysis- the distribution of hydro-refugia under different climatic conditions; and 2) modelling the connectivity/isolation of hominin groups via Agent-Based simulation. The main, intriguing conclusion suggested by the authors is that shifts between arid and wet climate conditions translated into different degree of habitat connectivity. I think this is an interesting paper with some potentially relevant implications on human metapopulation structure and evolution.

Given that hydrological analyses is not within my domain of expertise, I will focus my review on aspects pertaining the computational model proposed. The authors used a published agent-based model (Pathway Analysis Through Habitat, PATH), designed to detect patterns of successful inter-habitat migration. The key assumptions of PATH is that agents move stochastically between patches but have limited mobility range (dictated by environmental and biological constraints). Here patches are represented by the hydro-refugia detected by the hydrological analysis, the environmental constraints by the topography, and the biological constraint by presumed physiological limits (e.g. number of days of travel without water). Based on the simulation output the authors suggest an alternation between periods of high connectivity (i.e. most patches connected by migration routes) and periods with lower connectivity when the dispersal of agents are limited to local clusters of hydro-refugia.

While I think that the overall conclusion of the model is relevant and robust, I do have a number of comments, which I summarise in four points

1. Is an ABM really necessary in this case?

The key objective of PATH is to determine where we should expect to observe migration routes, definitely a better alternative to traditional GIS-based least-cost pathway (that offers only connectors rather than corridors) but somewhat very similar to other more widely used methods

based on circuit theory (e.g. McRae et al 2008, *Ecology* 10: 2712-2724.). From what I understand here, the primary objective is here to determine whether the hydro-refugia are connected or disconnected into multiple isolated clusters. If that is the case, I wonder why alternative (more common) methods were not considered. For example, one could generate a matrix defining the time required to reach from a patch *i* any other patch *j*, using the topographic constraints presented here, then determine which pairs can be "linked" given the physiological assumptions. The resulting matrix can then be used to create a simple graph that should be comparable to the output of PATH. The only difference I see is that PATH can generate a network with weighted edges recording the number of successful in migration and out migration from each patch. Perhaps I am missing something, but I would like to know why this specific approach has been used to measure patch connectivity, given its computationally less efficient, it's subject to stochastic components that are not explored, and have a higher number of parameters.

The referee is correct to point out that alternative methods for addressing the research objectives are available, including the use of circuit models. The specific advantages of using an ABM in this context are summarized by Bonabeau (2002, *PNAS*, 99, 7280-7287), in what is the most widely cited review of the use of ABMs to study human movement. Here, the author identifies three particular benefits of ABMs compared to other modelling approaches, namely: (i) ABM captures emergent phenomena; (ii) ABM provides a natural description of a system; and (iii) ABM is relatively flexible. In our study, connectivity between patches essentially arises as an emergent property of the simulation of the system's constituent units (the agents) and their interactions, capturing emergence from the bottom up when the simulation is run. As noted by Bonabeau (2002), ABMs particularly have value when there is potential for emergent phenomena, for example when individual behaviour is nonlinear and can be characterized by thresholds, or when individual behaviour exhibits memory, path-dependence, or temporal correlations, including learning and adaptation. While we do not explicitly model learning and adaptation in the current study, this is an aspect that is very relevant to hominin behaviour and evolution, which we are keen to explore in future research. One of the factors that influenced our decision to employ an ABM in the current investigation was to provide a basis for this future research. However, the points about flexibility, emergent properties and the relatively natural description of the system are applicable to our current study. In the context of the more natural description, as noted by Bonabeau, ABMs have a particular advantage when the behaviour of individuals is complex and cannot be easily defined through aggregate transition rates, and where activities are a more natural way of describing the system than processes – points that we believe apply to our current work. We would also add that the stochasticity, which the referee also refers to, we also see as an advantage of working with ABMs, as it offers a means of exploring uncertainty.

In terms of circuit models, as pointed about by McRae et al. 2008 (*Ecology* 10: 2712-2724), these have a number of limitations: they are restricted to Markovian random walks, and cannot incorporate correlated random walks, changes in movement behaviour with time, or mortality rates that increase with an organism's age. These are all elements that could be incorporated in the ABM, even though they are not all included in the model as it currently stands. In circuit models, random walkers can retrace their steps over and over, inflating mortality rates because travel time and exposure to mortality risks are increased, a limitation that our ABM avoided. Again as noted by McRae et al. (2008), individual-based models such as the ABM we employed offer much more flexibility than analytic models, can incorporate subtle effects of dispersal behaviour and other aspects of life history, and can simulate transient effects of landscape characteristics that evolve over time. These authors suggest that circuit models fill a niche

between simpler Euclidean or least-cost path analyses and more powerful analytic and simulation approaches such as ABMs.

2. Lack of quantitative measures of connectedness

The authors explore the simulation output purely in visual terms. While the results shown on figure 3 is undoubtedly compelling, it would help to quantify the level of patch connectivity in numerical terms. For connected graphs this could be something relatively simple (e.g. network diameter, node connectivity, link connectivity) while for disconnected graph even reporting the number of disconnected subgraphs might be helpful. Alternatively, one could benefit from the PATH model and record dispersal success, search time, or cell immigration (e.g. Tischendorf and Fahrig 2000, *Landscape Ecology*, 15, 633-641). In any case, there seems to be a wide range of options (see Kindlmann and Burel 2008, *Landscape Ecology*, 23, 879-890 for review) and I think a quantifiable description would allow a more formal parameter exploration and sensitivity analysis.

The referee is correct in pointing this out and we have addressed this directly. Outputs from the PATH model have now been analysed in three ways: (1) visually maps of successful agent journey are exported from Netlogo as raster files and uploaded for visualisation and analysis in ArcMap; (2) within ArcMap the PATH out can be converted into a binary landcover map (crossed or not crossed) for analysis in Fragstats which produces a range of connectivity variable; and (3) in addition the PATH model provides a matrix of successful journeys between named habitat (water) patches which can be exported and plotted within Matlab. We report this data for ten model runs per scenario and data for various sensitivity analyses in the SI and demonstrate clearly the statistical significance of the differences across the four scenarios models. We are particularly grateful to the reviewer for stimulating us in this direction rather than simply focusing on the geographical patterns.

3. Modelling pedestrian movement

The authors create a cost layer using Naismith's Rule, which converts terrain properties into walking speed. Thus the same "as the crow flies" distance would take a different amount of time depending on the terrain properties. But then they refer to their model parameters in km (line 325) rather than in hours. The authors are bit vague on this but I understand their assumption is a maximum of three days (line 314 "a maximum travel time of 3 days without water), with 10 hours spent on walking each (lines 321-322 "[...] assume that daily walking distance [...] over a 10-hour period). Now suppose we have two patches with an inter-distance of 120km but with a very rough terrain in the middle that requires more than 30 hours of travel time. Would the agent manage to reach its destination? I might have missed something, but the authors need to clarify how this aspect of the model has been implemented. Are they using a 30-hours limit or are they using specific distances (e.g.150km) but ignoring the amount of time taken, or are they using the smallest between the two? Without the source code (see next section) it is hard to understand how this was implemented.

The way this is implemented is via a simple 'energy quota'. We give each starting agent an energy value of 150. If the cost is zero (i.e. terrain flat) then they can travel 150 units (or cells) since each model cell is 1x1 km this is equivalent to 150 km. However, each cell has a cost related to the topography which consumes the agent's energy proportionately (i.e. rough terrain is more costly to traverse) this reduces the agents energy reserve and therefore the number of cells (i.e. effective distance) in total that can be traversed before death. Time is not modelled directly, but to work out the scaling we have to consider how far an individual could walk

theoretically in three days hence the ten hours used in the assumption. In the sensitivity analysis we have varied the starting energy from 120 through to 180 and the patterns observed remain robust. Travel distance greater than 180 km in three days are we believe unrealistic and while travel distance could be much less than 120 km selecting this as the lower limit means that our results favour connectivity and are therefore conservative with respect to looking at the potential for agent isolation. We have adjusted the text accordingly to make this more explicit.

A second minor remark is the assumption of a 30 hours walking time without water. I wonder whether it is right to assume a constant performance in physical activity without any water intake. I think a better way to approach this (and overcome possible issues/critiques with the model assumptions) is to identify the range of conditions (parameter settings) that can generate a specific outcome (e.g. a certain degree of patch connectivity). An example of this kind of approach is used by Gallagher et al 2015, PNAS, 112, 14218-14223.

We understand where the reviewer is coming from with this but our specific aim is to look at dispersal versus isolation – we don't know what patch connectivity should be? That is the point of the analysis.

4. Model Source Code

I could not find the original code of PATH developed by Hargrove and Westervelt, nor did the authors provided their implementation. There are certainly parameter settings (e.g. the number of agents, the stopping time, etc.) that are not crucial but nonetheless affect the outcome of the simulation. In order to make their work reproducible (and clarify any issues such as the one mentioned above on the pedestrian movements) I think it is paramount that the source code is shared, along with the specifics of the experiment design on an online repository (e.g. github, openABM) or in the Supplementary Information.

The PATH model is already in the public domain and can be obtained from: <http://extras.springer.com/2012/978-1-4614-1256-4>. We will also upload a copy of the code as part of the SI *should* the paper be accepted for publication.

Reviewers' Comments:

Reviewer #1 (Remarks to the Author)

The article has been improved, and the responses to the questions have been helpful in clarifying their approach. I am sympathetic to their view that groundwater fed springs are likely to have offered a highly resilient water source during times of prolonged drought, but have not been convinced from the evidence they have presented. There are several flaws that considerably weaken their argument. However, it is possibly that by addressing these flaws, being more open about the limitations of their assumptions and bringing in additional evidence to support the modelling results that a paper could be constructed that was more convincing. Below are my detailed responses to their responses to Reviewer 1. Where possibly I have offered suggestions for improvement

Question 1a: Mapping of springs. The caveat is acknowledged, however the authors have not addressed the main comments that under present day hydrological conditions (which the authors state they are using as analogy to drier historic conditions) there are many more springs in the wetter highland areas than in the rift floor. This makes their Figure 2 and associated discussion misleading and considerably weakens other more valid aspects of the paper. There are currently more springs in the Ethiopia escarpment and highlands than in the rift floor (see Ethiopian national database on water points, or Table 1 in Tucker et al 2014, Calow et al. 2009) . It could be that the maps used have considered only thermal springs, rather than cold water springs. A possibly way forward this is to explicitly state that the analysis only maps springs in the semi-arid and arid areas, and that springs in the more humid areas are too numerous to count and in essence accounted for in the areas mapped as having perennial rivers.

Question 1b – I am glad the authors recognise that climate is important – and it would have been better if they had made this clearer in the abstract (line 11). The point here is not of the basic groundwater principal that aquifer response time is related to aquifer diffusivity and geometry, but that climate has had to have driven groundwater recharge at some point – its just a question of when. The behaviour of the Nubian aquifer that the authors refer to is explained by its response to the cessation of active recharge (see Gossel et al. 2004 Hydrogeology Journal). Similarly for their own study. It may be useful for the authors to frame their discussion in the resilience literature, or groundwater and drought literature (Calow 1997, 2010) which includes long term recharge and aquifer response time, or to just state “short term climate fluctuations”, rather than “climate”

Question 2. I understand the authors' point that including people's current response to drought is questionable. However, people's stories of how their springs have responded to drought are still valid. I would suggest they draw on any existing research in this area.

Question 3. I understand the author's attraction to using a global dataset, but there has been much research and mapping of parts of the EAR that shows that the aquifers are highly complex and fractured and this directly impacts on the spring response. Kebede 2012 – Groundwater in Ethiopia, Alemayehu 2006 - Groundwater occurrence in Ethiopia ; Hydrogeological Map of Ethiopia; Hydrogeological Map of Tanzania. The porosity values of Gleeson et al. are likely to be on the high side, however, more important is the high degree of anisotropy, which coupled with topography means that much of the groundwater discharges in the highlands rather than making to all the way to the rift floor. The Swanson paper they refer to is not relevant here, since it refers to the situation where there is a large groundwater reservoir with a highly permeable layer which can transport groundwater to the spring. The volcanic rocks flanking the rift comprises multiple stacks of this layered system, allowing many opportunities for springs on the flanks.

Question 4. Please look at these papers below as a way to begin to validate your conceptual

model. As you'll see from these papers (there are others) this is an active areas of research and contentious. Here is a summary is that some recharge from the flanks does reach the rift floor – (maybe < 35% gets to the rift floor); Meteoric rainfall also important to recharging springs on the rift floor. Groundwater different form lakes – residence times can be 2000 years. I think these papers can be used to substantiate the hypothesis that groundwater is an important source of drinking water in dry periods, and reference to these will help bolster your arguments with an independent check on the simple modelling

Kebede et al. Groundwater origin and flow along selected transects in Ethiopian rift volcanic aquifers HJ 10.1007/s10040-007-0210-0

Bretzier et al. 2011 Groundwater origin and flow dynamics in active rift systems – A multi-isotope approach in the Main Ethiopian Rift JoH <http://dx.doi.org/10.1016/j.jhydrol.2011.03.022>

Rango 2010 The dynamics of central Main Ethiopian Rift waters: Evidence from δD , $\delta^{18}O$ and $87Sr/86Sr$ ratios. Applied Geochemistry <http://dx.doi.org/10.1016/j.apgeochem.2010.10.001>

In conclusion I still cannot support publication in its current form - and believe that additional work is required to present a convincing argument.

Reviewer #2 (Remarks to the Author)

The revised manuscript reads much better and I appreciate the better focus on the core message of the paper rather than evolutionary scenarios. I especially like Figure 7, which -- if it had a detailed caption -- would further explain why there is dispersal and gene flow at certain times. Therefore, in my opinion, Figure 7 needs a caption detailing what is occurring with the hominins at that point -- yes, I can figure it out but a couple of short sentences would make it easier, e.g., in a wet time periods hominins would be able to move across (E-W) and along the rift (N-S), as well as outside of the rift (?), due to the placement of the various types of water. Etc.

Otherwise the genetic and dispersal text as revised is good.

Reviewer #3 (Remarks to the Author)

This manuscript is much improved over the first version that I reviewed, and it is gratifying to see the care with which the authors addressed my concerns and those of other reviewers. The amount of work represented in this treatment of hydro-refugia is impressive and informative and will raise awareness of the potential importance of the distribution of water sources in human evolution. The manuscript is publishable with some additional revisions and wording suggestions, which I have indicated in the sticky notes on the attached. One remaining substantive issue relates to rainfall seasonality, which releases animals from depending on localized water sources one or twice every year (see below).

In general, the authors have exercised appropriate caution regarding claims about the importance of hydro-refugia to hominin evolution. There is one problem with their reasoning, however, the importance of seasonality and its cyclical impact on hominins' ability to move about the EARS landscape. Even during the arid precessional phases there would have been wet seasons in much of the EARS, otherwise there would be no primary productivity, no food, no survival for hominins or other animals. During the wet seasons (2 annually in some EARS regions), animals can and do disperse widely today (10's to 100's of km), then contract around water+food sources during the dry season(s). Springs and other hydro-refugia would be important resources during the dry seasons, but obviously much less important during the wet seasons. Population movements and gene flow thus would be controlled primarily by dispersal during wet seasons rather than during the dry seasons when hydro-refugia became critical.

The authors need to take this seasonality into account. Even during arid parts of precessional cycles, there would be periods of rain that keep the ecosystems going and allow the hominins and other organisms to disperse. Hydro-refugia would still be important during the dry parts of the

annual cycle, and even moreso during arid phases of the precessional cycle, but not so critical a controller of movements as proposed in the manuscript (year-around implied if not stated). Their general conclusions are still valid and support the proposal that other factors besides climate should be considered as drivers of hominin evolution, but I think the issue of seasonality must be addressed, even though it somewhat lessens/moderates the potential impact of hydro-refugia.

I also suggest a more honest compelling title, e.g., "Modeling the role of hydro-refugia in East African hominin evolution" or "Hydro-refugia as an important environmental driver in hominin evolution" (It's always good to have the first word of the title match the major theme of the paper.) The present title implies that you are analyzing evidence from the geological and paleontological record, which is not the case.

This paper will contribute important new ideas to debates about environmental drivers in hominin evolution.

Reviewer #4 (Remarks to the Author)

I had the opportunity to read the revised version of the manuscript and the response of the authors. They have answered most of my enquiries (the quantitative analysis of the connectedness, description of the pedestrian movement; and the sharing of the source code), but I am still not entirely convinced by their justification concerning the use of ABM in their work, and the related issue of their parameter choice.

ABM can indeed capture emergent phenomena, provide a natural description of a system, and offers a flexibility that is unavailable in most equation based models. These are certainly valid points and I am completely on board with the authors. My concern is that in this context, ABM is not used to build a heuristic model of human dispersal, nor "directly" to assess a specific historical hypothesis' it is used as an alternative way to "measure" potential connectedness of hydro-refugia. Two locations are connected (or not) depending on fixed properties (e.g. their geographic locations), how the ABM describes movement -which I understand from the source code it is a random walk with a fixed, unit length steps-, the model parameters (energy), and the stochasticity of the simulation. Each of these points should be explored thoroughly choices justified, as the output will clearly be affected. Now, to me, the simplest way to describe what kind of network we should expect to observe given a series of nodes is to define the edges using some measure of distance (e.g. Euclidean, cost distance, etc.) and see how the network structure changes as we change our threshold. In such scenario, we can conduct a sensitivity analysis showing for example that even if the estimate of our threshold (e.g. a walking distance of 20 hours instead of 30) is incorrect, the network structure (and my conclusions) would qualitatively remain the same.

With ABM we would need to re-run the model for each parameter setting in order to conduct a sensitivity analysis. This is fine, but of course begs the question on what values to explore, with the most viable option requiring the execution of at least extreme values (i.e. the lowest and the highest threshold). The authors do this, exploring, for example, three different travel distances of 120, 150, and 180. This is where I find an issue. The author states that (lines 400-403): "one can assume that daily walking distance for a modern human is likely to vary between 40 and 50 km over a 10 hour period" and that "[T]aking the least conservative value, the model scenarios were run with a three day distance of 150 km representing the maximum possible travel distance". As I mentioned in the previous comment, this estimates looks too high to me as it assumes that an individual is capable to keep up a daily range of 50km for three consecutive days without any water intake. More in general, my point is that we do not know what is the "correct" parameter value. Perhaps 150 km is spot-on, but perhaps is a huge overestimate, and we should consider something as low as 80km or less. If the authors can show that the results do not change even when we have a figure as low as, say 80, then that's a great news. If the authors can provide a better justification of their parameter range that's equally ok. But I would like to see more on this front. A larger and better-justified parameter exploration that ensures the robusticity of their

outcome would make this paper stronger. I think this is an extremely interesting work, and it would be a shame if, once published, its conclusions are questioned for the choice of a parameter setting!

Also, the code provided by the authors shows clearly that there are other parameters (i.e. “#travelers” and “max-turn-angle”), what parameter settings have been used for these and why? Also, when was each simulation stopped? Was this based on certain conditions being met (e.g. all agents having zero energy) or a number of time-steps? In the latter case was this based on the stationarity of the output, and if so how was this measured? These are key aspects required to make the study reproducible and must be included and discussed at least in the supplementary materials.

Response to Reviewers' Comments (Second Revision)

(Line numbers refer to the related MS version uploaded which includes 'tracked changes')

Reviewer #1 (Remarks to the Author):

The article has been improved, and the responses to the questions have been helpful in clarifying their approach. I am sympathetic to their view that groundwater fed springs are likely to have offered a highly resilient water source during times of prolonged drought, but have not been convinced from the evidence they have presented. There are several flaws that considerably weaken their argument. However, it is possibly that by addressing these flaws, being more open about the limitations of their assumptions and bringing in additional evidence to support the modelling results that a paper could be constructed that was more convincing. Below are my detailed responses to their responses to Reviewer 1. Where possibly I have offered suggestions for improvement

Thanks to the Reviewer for their positive suggestions for improving the manuscript.

Question 1a: Mapping of springs. The caveat is acknowledged, however the authors have not addressed the main comments that under present day hydrological conditions (which the authors state they are using as analogy to drier historic conditions) there are many more springs in the wetter highland areas than in the rift floor. This makes their Figure 2 and associated discussion misleading and considerably weakens other more valid aspects of the paper. There are currently more springs in the Ethiopia escarpment and highlands than in the rift floor (see Ethiopian national database on water points, or Table 1 in Tucker et al 2014, Calow et al. 2009) . It could be that the maps used have considered only thermal springs, rather than cold water springs. A possibly way forward this is to explicitly state that the analysis only maps springs in the semi-arid and arid areas, and that springs in the more humid areas are too numerous to count and in essence accounted for in the areas mapped as having perennial rivers.

The mapping did distinguish geothermal from cold springs and we have treated these types of springs differently as stated in the Methods section. However, we agree that the Reviewer's suggestion here is a sensible one, and have now added further caveats to the text regarding the likely under-mapping of springs in the more humid parts of the study area in Lines 65-66 and 268-270. We have amended Figure 2 to exclude the most humid areas (>60 mm/y recharge) for this reason, and have also added a more nuanced discussion in lines 78-86 based on the new figure. This indicates a slight bias towards a smaller total number of springs in the most arid areas, but also a counter bias in the presence of the most persistence springs. The Reviewer's persistence is appreciated on this point as it has certainly led to a more robust presentation of the evidence on this issue.

Question 1b – I am glad the authors recognise that climate is important – and it would have been better if they had made this clearer in the abstract (line 11). The point here is not of the basic groundwater principal that aquifer response time is related to aquifer diffusivity and geometry, but that climate has had to have driven groundwater recharge at some point – its just a question of when. The behaviour of the Nubian aquifer that the authors refer to is explained by its response to the cessation of active recharge (see Gossel et al. 2004 Hydrogeology Journal). Similarly for their own study. It may be useful for the authors to frame their discussion in the resilience literature, or groundwater and drought literature

(Calow 1997, 2010) which includes long term recharge and aquifer response time, or to just state “short term climate fluctuations”, rather than “climate”

We agree with the Reviewer, and we have clarified the text slightly on line 100 to make this even clearer, as well as amended the abstract in Lines 11-14 to make it clearer that the spatially variable groundwater response times control the ‘buffering’ of the climate signal.

This ‘basic groundwater principle’ may seem basic to the Reviewer (and indeed also to the hydrogeologists among the authors of the paper), but it is not yet widely appreciated among the interdisciplinary communities interested in paleohydrology, and the paleoecology of hominins in particular. We believe that communicating this fundamental groundwater principle, to a much wider audience, is therefore a very important contribution that this paper can make.

Question 2. I understand the authors’ point that including people’s current response to drought is questionable. However, people’s stories of how their springs have responded to drought are still valid. I would suggest they draw on any existing research in this area.

Thanks for the Reviewer’s suggestion. We haven’t been able to find any peer reviewed literature which specifically helps much here in terms of longer term ‘indigenous knowledge’. However, we have added, in Lines 200-202, citations to Calow et al 2010 and Tucker et al 2014 which are useful and relevant cases in point for the experience of present day Ethiopia.

Question 3. I understand the author’s attraction to using a global dataset, but there has been much research and mapping of parts of the EAR that shows that the aquifers are highly complex and fractured and this directly impacts on the spring response. Kebede 2012 – Groundwater in Ethiopia, Alemayehu 2006 - Groundwater occurrence in Ethiopia ; Hydrogeological Map of Ethiopia; Hydrogeological Map of Tanzania. The porosity values of Gleeson et al. are likely to be on the high side, however, more important is the high degree of anisotropy, which coupled with topography means that much of the groundwater discharges in the highlands rather than making to all the way to the rift floor. The Swanson paper they refer to is not relevant here, since it refers to the situation where there is a large groundwater reservoir with a highly permeable layer which can transport groundwater to the spring. The volcanic rocks flanking the rift comprises multiple stacks of this layered system, allowing many opportunities for springs on the flanks.

We are aware of the research and mapping that the Reviewer refers to. In response, we first want to reiterate that while we have used a ‘global dataset’ it has the same geological basis as the best complete regional data, but has the added benefit of giving regionally consistent indications in aquifer properties as needed for the modeling. However, we have added an explicit acknowledgement in the text in Lines 327-330 that the parameterisation does not take account of local anisotropy and heterogeneity.

We agree that it is likely to be the case that in parts of the study area (in particular the setting that the Reviewer is referring to) that anisotropy will be an important control on the distribution of the springs. However, we are not using groundwater modelling with ‘bulk’ (and thus isotropic) parameters to infer spring positions, which would indeed be a spurious approach in such a context. Rather we are using actual observed (mapped) spring locations

and then assigning bulk aquifer parameters to assess the likely groundwater response times – this is an important conceptual difference.

That said, we agree that additional discussion on this is important with regard to the Reviewer's comment from their opening paragraph 'being more open about the limitations of their assumptions', and we have therefore now modified the text at various points in light of this comment.

Question 4. Please look at these papers below as a way to begin to validate your conceptual model. As you'll see from these papers (there are others) this is an active areas of research and contentious. Here is a summary is that some recharge from the flanks does reach the rift floor – (maybe < 35% gets to the rift floor); Meteoric rainfall also important to recharging springs on the rift floor. Groundwater different form lakes – residence times can be 2000 years. I think these papers can be used to substantiate the hypothesis that groundwater is an important source of drinking water in dry periods, and reference to these will help bolster your arguments with an independent check on the simple modelling

Kebede et al. Groundwater origin and flow along selected transects in Ethiopian rift volcanic aquifers HJ 10.1007/s10040-007-0210-0

Bretzier et al. 2011 Groundwater origin and flow dynamics in active rift systems – A multi-isotope approach in the Main Ethiopian Rift JoH

<http://dx.doi.org/10.1016/j.jhydrol.2011.03.022>

Rango 2010 The dynamics of central Main Ethiopian Rift waters: Evidence from δD , $\delta^{18}O$ and $87Sr/86Sr$ ratios. Applied Geochemistry

<http://dx.doi.org/10.1016/j.apgeochem.2010.10.001>

In response to the previous comments by Reviewer 1, we already carried out a substantial amount of extra work to test the numerical modelling approach against existing discharge data (see Methods section 'Model testing'). However, since there is a great paucity of such spring discharge data for the region, we agree that also using the existing geochemical evidence to additionally validate the approach is a good idea. Although the papers the Reviewer lists, and related literature, do indicate substantial complexity in the EARS hydrogeology, they also give a consistent picture of flow patterns with deeper older water on longer flow paths to the rift floor and shallower younger water in shorter flow paths on the rift flanks. This is consistent with the assumptions in the combined mapping-modelling approach we have used, so we have enriched the text with the insights from this literature.

In conclusion I still cannot support publication in its current form - and believe that additional work is required to present a convincing argument.

We appreciate the insights that this Reviewer has provided to help us tighten the groundwater aspects of the paper and present a more convincing argument. Given the work we have put in to take account of the comments from both phases of review, and the improvements in the analysis that have resulted, we hope the Reviewer will now be supportive of publication.

Reviewer #2 (Remarks to the Author):

The revised manuscript reads much better and I appreciate the better focus on the core message of the paper rather than evolutionary scenarios. I especially like Figure 7, which -- if it had a detailed caption -- would further explain why there is dispersal and gene flow at

certain times. Therefore, in my opinion, Figure 7 needs a caption detailing what is occurring with the hominins at that point -- yes, I can figure it out but a couple of short sentences would make it easier, e.g., in a wet time periods hominins would be able to move across (E-W) and along the rift (N-S), as well as outside of the rift (?), due to the placement of the various types of water. Etc.

Otherwise the genetic and dispersal text as revised is good.

Thanks to the Reviewer for this suggestion. We have now added a revised caption for Fig. 7 as follows:

“Figure 7: Conceptual model showing the role of springs across various climate scenarios. Under the dry scenario (right panel) hominin survival is focused on single springs (or clusters of springs) and movement between springs (or spring clusters) is limited. As climate improves (central panel) the availability of water sources increases particularly as the water table intersects rift-flank rivers. Springs high on the rift sides may act to link rift-flank rivers with water sources in the rift facilitating transverse rift movement. As water become widely available (left panel) hominin movement occurs in all direction including along the rift axis.”

Reviewer #3 (Remarks to the Author):

This manuscript is much improved over the first version that I reviewed, and it is gratifying to see the care with which the authors addressed my concerns and those of other reviewers. The amount of work represented in this treatment of hydro-refugia is impressive and informative and will raise awareness of the potential importance of the distribution of water sources in human evolution.

Thank you to the Reviewer for the appreciation of the work that has gone into this paper and its importance.

The manuscript is publishable with some additional revisions and wording suggestions, which I have indicated in the sticky notes on the attached. One remaining substantive issue relates to rainfall seasonality, which releases animals from depending on localized water sources one or twice every year (see below).

In general, the authors have exercised appropriate caution regarding claims about the importance of hydro-refugia to hominin evolution. There is one problem with their reasoning, however, the importance of seasonality and its cyclical impact on hominins' ability to move about the EARS landscape. Even during the arid precessional phases there would have been wet seasons in much of the EARS, otherwise there would be no primary productivity, no food, no survival for hominins or other animals. During the wet seasons (2 annually in some EARS regions), animals can and do disperse widely today (10's to 100's of km), then contract around water+food sources during the dry season(s). Springs and other hydro-refugia would be important resources during the dry seasons, but obviously much less important during the wet seasons. Population movements and gene flow thus would be controlled primarily by dispersal during wet seasons rather than during the dry seasons when hydro-refugia became critical.

The authors need to take this seasonality into account. Even during arid parts of precessional cycles, there would be periods of rain that keep the ecosystems going and allow the hominins

and other organisms to disperse. Hydro-refugia would still be important during the dry parts of the annual cycle, and even moreso during arid phases of the precessional cycle, but not so critical a controller of movements as proposed in the manuscript (year-around implied if not stated). Their general conclusions are still valid and support the proposal that other factors besides climate should be considered as drivers of hominin evolution, but I think the issue of seasonality must be addressed, even though it somewhat lessens/moderates the potential impact of hydro-refugia.

Many thanks to the Reviewer for the additional thoughts on how to improve the paper.

Although our analysis is primarily focussed on precessional variability, the Reviewer is absolutely correct that seasonality could add additional nuances on hominin mobility. We have added some discussion of this accordingly to the text in Lines 196-211 as follows:

“In addition to the long timescale changes in climate expected through a precessional cycle, shorter term variations (e.g. seasonal dry periods or multi-year droughts) would have altered the availability of fresh water. For the ‘present’ scenarios, the effect of seasonality on the potential connectivity of hydro-refugia is incorporated in the analysis by comparing the ‘present wet’ and ‘present dry’ scenarios (Figures 5b and S10) . Such patterns of variation in the location of available spring water are in accordance with the modern experience of East African communities^{29,30}. For the ‘future wet’ scenario, mobility is already so easy that seasonality would make little difference (Fig. 5c). For the future ‘dry’ scenario, the impact of seasonality is harder to constrain but in the driest parts of the precessional cycle envisaged, seasonal expansion of the drainage network is likely to have been much less than that during the present day. This is because both runoff and recharge are strongly controlled by antecedent moisture and water table conditions^{21,31}. Hence while seasonal mobility may be enhanced to some extent even during short wet periods, a sustained drier prevailing climate would result in decreased streamflow and a less expansive stream network than observed in the present day even during periods of relative (e.g. seasonal) wetness.”

I also suggest a more honest compelling title, e.g., "Modeling the role of hydro-refugia in East African hominin evolution" or "Hydro-refugia as an important environmental driver in hominin evolution" (It's always good to have the first word of the title match the major theme of the paper.) The present title implies that you are analyzing evidence from the geological and paleontological record, which is not the case.

We have amended the title to reflect the modelling focus as follows:

“Modelling the role of groundwater hydro-refugia in East African hominin evolution and dispersal”

This paper will contribute important new ideas to debates about environmental drivers in hominin evolution.

We hope so!

Line 19 during the dry season. See later comments about the importance of seasonality - many animals disperse widely during the wet season(s), so the importance of hydro-refugia would have been on a seasonal cycle as well. It is possible that hydro-refugia, and the ability to mentally map them, allowed hominins to disperse during the dry as well as the wet

seasons, when water would not have been a limiting factor. Other animals can also learn where such resources are and how to find them, so the networks of hydro-refugia wouldn't just affect hominin movements

See reply to this point above.

Our replies to comments on the annotated pdf the Reviewer attached are as follows:

Line 39 Not THE only key resource, but one of several. This is overstated.

We have the text amended to “a key resource”.

Line 40 You must say what you mean in terms of time scale. Many of the lakes and rivers are quite persistent on ecological time scales.

We have amended the text to indicate that we are talking about ‘greater than seasonal’ timescales.

Line 42 Conjecture. It is true that the lakes would have fluctuated, but we don't know how dried out they were during the swings of the precessional cycle.

We have replaced ‘would have’ with ‘thought to have’.

Line 44 Overstated.

See reply to next comment.

Line 45 But if 30% of the EARS still has surface water, then this is not necessarily analogous to the "driest parts" of the cycles.

We have amended ‘driest’ to ‘dry’ to be consistent with the ‘drier’ than present model scenario we present later in the paper.

Line 56 How is "significant" defined?

What we mean by ‘significant’ here is defined in the Methods but we have now reiterated it in the main text to make it clearer: “enough running water to provide drinking requirements for 100s of animals and to sustain a small wetland (see Methods)”.

Line 59 Give the number, e.g., 10 out of X, and state why the catchments draining the Ethiopian Highlands are excluded, because these obviously would have been important persistent water sources for hominins in the EARS.

We have amended the next sentence to make this clearer: “8 out of 34”. We haven't excluded the Ethiopian Highlands from that statistic or the modelling, but rather are just stating here that it is the wetter areas such as this that can sustain freshwater lakes downstream despite local aridity in the vicinity of the lake.

Line 65 But the water is not potable for humans on any sustained basis.

We have amended the text as suggested.

Line 137 But higher elevations should continue to have higher rainfall even through the drier part of the cycle, therefore spring frequency should increase with elevation through both the drier or wetter parts of the cycle. What will change is the volume of the spring flow and how far it can go downhill.

We agree with the Reviewer's conjecture that a topographic rainfall gradient is likely to persist, while the relative changes between highlands and lowlands are unknown. However, even the proportional decrease in the regional rainfall we have assumed would still lead to the general trend we are describing. We have added a caveat in the text to this effect and clarified our stated assumptions, which we feel is a transparent way to approach this uncertainty.

Line 149 There is a flaw in this reasoning - the importance of seasonality. Even during the arid phases there must still have been wet seasons, otherwise there would be no primary productivity, no food, no survival for hominins or other animals. During the wet season, animals can and do disperse widely today (10's to 100's of km), then contract around water+food sources during the dry season. (Hippos are known to wander far from water sources during wet seasons, and of course there is the iconic migration of wildebeest that follow seasonal cycles. Springs and other hydro-refugia would be important resources during the dry seasons, but obviously much less important during the wet seasons. Gene flow thus would more likely be controlled by dispersal during the wet seasons rather than during the dry seasons when hydro-refugia became critical.

You need to take seasonality into account, and the fact that even during arid parts of precessional cycles, there would be times of rain that keep the ecosystems going and allow the hominins and other organisms to disperse. Hydro-refugia would still be important during the dry parts of the annual cycle.

See reply to this point above.

Reviewer #4 (Remarks to the Author):

I had the opportunity to read the revised version of the manuscript and the response of the authors. They have answered most of my enquiries (the quantitative analysis of the connectedness, description of the pedestrian movement; and the sharing of the source code), but I am still not entirely convinced by their justification concerning the use of ABM in their work, and the related issue of their parameter choice.

ABM can indeed capture emergent phenomena, provide a natural description of a system, and offers a flexibility that is unavailable in most equation based models. These are certainly valid points and I am completely on board with the authors. My concern is that in this context, ABM is not used to build a heuristic model of human dispersal, nor "directly" to assess a specific historical hypothesis' it is used as an alternative way to "measure" potential connectedness of hydro-refugia. Two locations are connected (or not) depending on fixed properties (e.g. their geographic locations), how the ABM describes movement -which I understand from the source code it is a random walk with a fixed, unit length steps-, the model parameters (energy), and the stochasticity of the simulation. Each of these points should be explored thoroughly choices justified, as the output will clearly be affected. Now,

to me, the simplest way to describe what kind of network we should expect to observe given a series of nodes is to define the edges using some measure of distance (e.g. Euclidean, cost distance, etc.) and see how the network structure changes as we change our threshold. In such scenario, we can conduct a sensitivity analysis showing for example that even if the estimate of our threshold (e.g. a walking distance of 20 hours instead of 30) is incorrect, the network structure (and my conclusions) would qualitatively remain the same.

With ABM we would need to re-run the model for each parameter setting in order to conduct a sensitivity analysis. This is fine, but of course begs the question on what values to explore, with the most viable option requiring the execution of at least extreme values (i.e. the lowest and the highest threshold). The authors do this, exploring, for example, three different travel distances of 120, 150, and 180. This is where I find an issue. The author states that (lines 400-403): “one can assume that daily walking distance for a modern human is likely to vary between 40 and 50 km over a 10 hour period” and that “[T]aking the least conservative value, the model scenarios were run with a three day distance of 150 km representing the maximum possible travel distance”. As I mentioned in the previous comment, this estimates looks too high to me as it assumes that an individual is capable to keep up a daily range of 50km for three consecutive days without any water intake.

More in general, my point is that we do not know what is the “correct” parameter value. Perhaps 150 km is spot-on, but perhaps is a huge overestimate, and we should consider something as low as 80km or less. If the authors can show that the results do not change even when we have a figure as low as, say 80, then that’s a great news. If the authors can provide a better justification of their parameter range that’s equally ok. But I would like to see more on this front. A larger and better-justified parameter exploration that ensures the robusticity of their outcome would make this paper stronger. I think this is an extremely interesting work, and it would be a shame if, once published, its conclusions are questioned for the choice of a parameter setting!

Also, the code provided by the authors shows clearly that there are other parameters (i.e. “#travelers” and “max-turn-angle”), what parameter settings have been used for these and why? Also, when was each simulation stopped? Was this based on certain conditions being met (e.g. all agents having zero energy) or a number of time-steps? In the latter case was this based on the stationarity of the output, and if so how was this measured? These are key aspects required to make the study reproducible and must be included and discussed at least in the supplementary materials.

Thanks again for this Reviewer’s input - we are delighted that he/she is ‘completely on board with the authors’ in our use of an ABM. However we disagree that the ‘ABM is not used to build a heuristic model of human dispersal, nor “directly” to assess a specific historical hypothesis’. We would refer to the Reviewer to our previous response and suggest gently that clearly he/she would approach the same problem from a different perspective. No two researchers would approach the same problem in exactly the same manner. We therefore believe our approach is equally valid, even though the Reviewer would have approached it differently. In our study, connectivity between patches essentially arises as an *emergent* property of the simulation of the system’s constituent units (the agents) and their interactions, capturing emergence from the bottom up when the simulation is run. Connectivity is the result of emergent behaviour within our model.

The Reviewer states that “Two locations are connected (or not) depending on fixed properties (e.g. their geographic locations).” But in fact there is a level of complexity incorporated in our model that is not fully reflected in this statement. The landscape is heterogeneous in terms of land cover, slope, roughness and this affects the ability of hominins to traverse an area. This we have captured in our model, explicitly by incorporating variation in transit time in response to slope and roughness. There are many potential routes between two points in a landscape, and our model allows agents to explore a wide range of different routes, according to the decisions that would be made by an individual walking across a landscape. Only those successful transits between water bodies are considered in our analysis as evidence of linkage between “nodes” (or water bodies). So the ABM approach is heuristic, in this sense, and captures a much more complex interpretation of what constitutes a connection than would be achieved solely by a network analysis; the latter would not permit landscape heterogeneity to be incorporated as we have been able to do with the ABM

We appreciate fully that Reviewer #4 believes that our total travel distance of 150 km may be a little large. This is open to debate, not least because of the unknowable component of ‘urgency’ a hominin would face without water. San Bushmen can travel huge distances if required although routinely they choose more conservative home ranges. If we use the total travel distance preferred by the Reviewer of 80 km in three days our overall conclusions remain intact, that: (1) single springs act as hydrorefugia in dry periods; (2) that movement between closely spaced springs is still possible; (3) that as the climate ameliorates connectivity between the rift floor and flank rivers is aided by springs on the rift sides; and (4) under the wettest of conditions movement is unlimited. These conclusions do not change with the travel distance, simply the point at which these elements occur along the climate continuum we have explored.

We accept that we could do more to justify our parameter selection and have added a few additional points to the manuscript in light of this. While the issue of the ‘#travellers’ and ‘turn angle’ is dealt with in the manuscript we clearly need to strengthen this and have accordingly made modifications to the Methods section.

Reviewer #1:

Remarks to the Author:

Thanks for the detailed responses to my queries. The paper is considerably strengthened – below are my detailed responses.

Q1a – happy with the response and it is now clearer.

Q1b Glad this has been taken on board and the modification to the abstract and title is good. However, need to make sure this is carried through the document. For example 74 – 81 is clumsily written and I'm sure you will want the opportunity to write this in a clearer fashion. The first sentence is redundant (and confusing). Also I don't think you can see from Figure 2 that there are less springs in the arid areas. 30% of them are in areas with <10 mm recharge. Much better to say that "Springs exist in all the areas and that modelling indicates that those in the drier areas are the most persistent. This is explained by"

Also will need to modify Figure 2 title. If you need to mention climate, then it should be qualified as modern climate.

Q2 – Ok – Reference to these two spring studies is sufficient. There are some interesting studies looking at spring behaviour during the current Ethiopian drought – but not published yet.

Q3 – OK = acknowledging the models limitations is sufficient.

Q4 – Glad you have found the additional literature helpful. Adding in this independent verification of the groundwater flow in the EAR strengthens the paper.

Therefore happy that this paper proceeds to publication, subject to some minor edits as outlined above.

Alan MacDonald

Reviewer #3:

Remarks to the Author:

I am very happy to find this manuscript much improved over previous versions that I have reviewed. The authors have taken the critical comments, suggestions, and edits and done a very good job of rectifying the problems and providing cautionary or qualified statements where necessary. This effort has transformed the writing into a highly professional, balanced, and convincing case for the importance of the geographic distribution of water resources (= hydro-refuges at the drier end of the climate spectrum). The modelling is now more clearly explained and the amount of careful consideration and testing of the different variables, as well as the quality of the figures, is clear and very impressive. Several of the figures in the Supplementary Materials are even more interesting and informative than those in the body of the paper - the extensive SM will be a valuable resource for careful readers.

This paper is a very significant contribution that will greatly enhance serious consideration of the importance of water sources in hominin evolution and how these could have moderated the impact of precession-driven and other cycles of climate change. It should generate a lot of discussion and perhaps controversy; the successive rounds of reviews and the authors' responses, which greatly improved communication of their findings exemplify the best outcome of a serious and positive peer review process.

Note that I have made a few suggestions and corrections in the pdf (yellow sticky notes) including the figures, and also a few notes on the SM pdf.

Reviewer #4:

Remarks to the Author:

I think the article improved and although I still have philosophical disagreement with the authors (but I don't think this is something that would ultimately influence the result of the analysis in qualitative terms) I do think this is a very interesting read.

My critique is not on the use of ABM per se, but simply the fact that I am not absolutely sure whether the additional insights provided by the ABM fully justify its computing cost and the necessity of continuous re-runs to re-assess parameter assumptions. That said I am relieved that the results are qualitatively unchanged when lower travel distance parameters are used. These are parameters that are hard, if not impossible, to estimate and the best we can do is to explore at least the extreme settings and figure out at what point the assumption starts to matter so much that the results become different. I am also happy with the additional descriptions of the model parameters.

Response to Reviewers' Comments (Third Revision)

Reviewer #1 (Remarks to the Author):

Thanks for the detailed responses to my queries. The paper is considerably strengthened – below are my detailed responses.

Many thanks to the Reviewer for all their input.

Q1a – happy with the response and it is now clearer.

Q1b Glad this has been taken on board and the modification to the abstract and title is good. However, need to make sure this is carried through the document. For example 74 – 81 is clumsily written and I'm sure you will want the opportunity to write this in a clearer fashion. The first sentence is redundant (and confusing). Also I don't think you can see from Figure 2 that there are less spring in the arid areas. 30% of them are in areas with <10 mm recharge. Much better to say that "Springs exist in all the areas and that modelling indicates that those in the drier areas are the most persistent. This is explained by"

Thanks for this suggestion. Yes, we are not talking here about there being fewer springs in arid areas, but rather fewer springs in comparison to what would be expected if springs were randomly distributed. We have now clarified the text to make this clearer as follows:

"Within the drier parts of the study area where spring mapping is most reliable, in comparison to what would be expected if springs were randomly distributed across the landscape, there is a slight bias for fewer springs to be located in the most arid areas (Fig. 2). However, ..."

Also will need to modify Figure 2 title.

We have also amended the Figure 2 legend by adding the following:

"The 'regional recharge rate' curve is indicative of the cumulative frequency distribution that would be expected if springs were randomly distributed across the landscape."

If you need to mention climate, then it should be qualified as modern climate.

Ok, we have done so.

Q2 – Ok – Reference to these two spring studies is sufficient. There are some interesting studies looking at spring behaviour during the current Ethiopian drought – but not published yet.

We look forward to reading about this in any forthcoming papers.

Q3 – OK = acknowledging the models limitations is sufficient.

Q4 – Glad you have found the additional literature helpful. Adding in this independent verification of the groundwater flow in the EAR strengthens the paper.

Therefore happy that this paper proceeds to publication, subject to some minor edits as outlined above.

Alan MacDonald

Reviewer #3 (Remarks to the Author):

I am very happy to find this manuscript much improved over previous versions that I have reviewed. The authors have taken the critical comments, suggestions, and edits and done a very good job of rectifying the problems and providing cautionary or qualified statements where necessary. This effort has transformed the writing into a highly professional, balanced, and convincing case for the importance of the geographic distribution of water resources (= hydro-refuges at the drier end of the climate spectrum). The modelling is now more clearly explained and the amount of careful consideration and testing of the different variables, as well as the quality of the figures, is clear and very impressive. Several of the figures in the Supplementary Materials are even more interesting and informative than those in the body of the paper - the extensive SM will be a valuable resource for careful readers.

This paper is a very significant contribution that will greatly enhance serious consideration of the importance of water sources in hominin evolution and how these could have moderated the impact of precession-driven and other cycles of climate change. It should generate a lot of discussion and perhaps controversy; the successive rounds of reviews and the authors' responses, which greatly improved communication of their findings exemplify the best outcome of a serious and positive peer review process.

We agree, many thanks to the Reviewer for all their input.

Note that I have made a few suggestions and corrections in the pdf (yellow sticky notes) including the figures, and also a few notes on the SM pdf.

Line 73 add comma here.

We have done so.

Line 93 "...discharge, greatly reduces the groundwater response time, and thus increases..."

We have made the change as suggested.

Line 97 "...all of which determine.."

We have made the change as suggested.

Line 130 I assume that you mean in the geological record - this should be clearer.

We have amended the sentence to make this clearer:

"The fact that at least some persistent springs are likely to have been present during past dry periods in areas mapped as having perennial streams in the present day also suggests our predictions are conservative (Fig. S4)."

Line 469 Also Fig. S6

A reference to Fig. S6 has now been added.

Line 493 Thermal stress of the terrain would also likely be important, e.g., black rocks vs. white sand vs. vegetated substrate, etc. Also, do we assume that travel occurred only by day?

We have now added a statement acknowledging that there may be other aspects of terrain which might impede movement such as vegetation type and surface albedo or 'going characteristics'. Yes, we have assumed movement only by day as stated a little earlier in the same methods section.

Figure 3 legend Caption should say that this is based on the model results.

We have made the change as suggested.

Figure 7 The left to right order here is different from Figure 5 - why not make them the same?

We have made the change as suggested.

Figure S12 Please provide more explanation for why the connecting (dark) lines are weaker in the intermediate scaling (2). Or are these out of order?

We have re-ordered the figure but note that the scaling order in the original was correct.

Figure S13 This is a very informative and interesting figure!

Figure S15 Please say a little more in the caption about what this means.

We have added a bit more explanation to the legend as follows:

“The 95% confidence ellipses show the significant level of difference between the three primary runs modelled.”

Reviewer #4 (Remarks to the Author):

I think the article improved and although I still have philosophical disagreement with the authors (but I don't think this is something that would ultimately influence the result of the analysis in qualitative terms) I do think this in a very interesting read.

My critique is not one the use of ABM per se, but simply the fact that I am not absolutely sure whether the additional insights provided by the ABM fully justifies its computing cost and the necessity of continuous re-runs to re-assess parameter assumptions. That said I am relieved that the results are qualitatively unchanged when lower travel distance parameter are used. These are parameters that are hard, if not impossible, to estimate and the best we can do is to explore at least the extreme settings and figure out at what point the assumption starts to matter so much that the results becomes different. I am also happy with the additional descriptions of the model parameters.

Many thanks to the Reviewer for all their input and willingness to ‘agree to disagree’ on this philosophical point.